

# Exploring precipitation pattern scaling methodologies and robustness among CMIP5 models

Ben Kravitz[1], Cary Lynch[2], Corinne Hartin[2], and Ben Bond-Lamberty[2]

[1]Atmospheric Sciences and Global Change Division, Pacific Northwest National Laboratory, Richland, WA, USA.
[2]Joint Global Change Research Institute, Pacific Northwest National Laboratory, College Park, MD, USA.

*Correspondence to:* Ben Kravitz, P.O. Box 999, MSIN K9-30, Richland, WA 99352, USA. (ben.kravitz@pnnl.gov)

**Abstract.** Pattern scaling is a well established method for approximating modeled spatial distributions of changes in temperature by assuming a time-invariant pattern that scales with changes in global mean temperature. We compare three methods of pattern scaling for precipitation (regression, epoch difference, and a physically-based method) and evaluate which methods are "better" in particular circumstances by quantifying their robustness to interpolation/extrapolation, inter-model variations, and inter-scenario variations. Although the regression and epoch difference methods (the two most commonly used methods of pattern scaling) have better absolute performance in reconstructing the climate model output by two orders of magnitude (measured as an area-weighted root mean square error), the physically-based method shows a greater degree of robustness (less relative root-mean-square variation than the other two methods) and could be a particularly advantageous method if outstanding biases could be reduced. We decompose the precipitation response in the RCP8.5 scenario into a $CO_2$ portion and a non-$CO_2$ portion; these two patterns oppose each other in sign. Due to low signal-to-noise ratios, extrapolating RCP8.5 patterns to reconstruct precipitation change in the RCP2.6 scenario results in double the error of reconstructing the RCP8.5 scenario for the regression and epoch difference methods. The methodologies discussed in this paper can help provide precipitation fields for other models (including integrated assessment models or impacts assessment models) for a wide variety of scenarios of future climate change.



# 1 Introduction

Quantifying uncertainties in projections of climate change is one of the cornerstone investigative areas in climate science. There are numerous sources of uncertainty, including parametric (which parameter values are the "right" ones), structural (which key processes are missing or poorly characterized), and scenario (how climate forcing agents will change in the future). One commonality among these sources is that uncertainties in each of them can be explored using climate models.

Earth System Models (ESMs) are the gold standard of climate models used for projections of global change, as they incorporate many of the fundamentally climatically important processes, including atmosphere, land, ocean, and sea ice responses and feedbacks, as well as interactions between these different areas. Their complexity means that these models are often computationally expensive, however, so any sensitivity studies or uncertainty quantification efforts using them are necessarily limited. No modern uncertainty quantification technique is capable of fully characterizing the space of ESM uncertainties and how they affect projections of climate change (Qian et al., 2016).

Emulators of ESMs are often an effective compromise for exploring uncertainty by sacrificing precision for vastly improved computational efficiency. This allows other models, such as integrated assessment models or impacts assessment models, to include a ESM-emulating climate component and incorporate feedbacks between the climate and other sectors. There are many methods of building emulators (see MacMartin and Kravitz, 2016, for a discussion of different linear, time-invariant approaches), but one of the most commonly used methods is pattern scaling, described in more detail in Section 2.1. This methodology involves computing a time-invariant pattern of change in a variable in response to change in global mean temperature, which vastly reduces the dimensionality of input needed to produce projections of climate change.

Pattern scaling has a long history of research (e.g., Mitchell, 2003) and has been shown to be reasonably accurate for a variety of purposes. Lynch et al. (2016) provide a review of pattern scaling of temperature, as well as an in-depth exploration of two commonly used pattern scaling methods (regression and epoch difference methods, described later in Section 2.1). Both of these methods perform quite well in reproducing the actual model output for temperature. Conversely, comparatively little work has been done on pattern scaling for precipitation. Ruosteenoja et al. (2007) found that local precipitation changes are generally linear with global mean temperature change, with errors of 15–30% over 90 years of simulation. Holden and Edwards (2010) identified the importance of covariance between local temperature change and local precipitation change, and Frieler et al. (2012) furthered this discovery, concluding that no single fit (e.g., regression coefficients) will be applicable to all grid points. Herger et al. (2015) used a novel method of piecing together results associated with the desired global mean temperature change and found excellent agreement with model output (errors rarely exceed $0.3 \ \mathrm{mm \ day^{-1}}$). In a different style of emulation, Castruccio et al. (2014) trained a statistical model on pre-computed climate model simulations and found that it was capable of capturing nonlinearities in the response in ways that pattern scaling inherently cannot. To the best of our knowledge, no previous study has compared different methods of pattern scaling of precipitation.

Here we provide a systematic (although non-exhaustive) assessment of the robustness of pattern scaling of precipitation. Section 3 focuses on pattern scaling the response to temperature changes solely due to carbon dioxide increases, looking at



interpolation, extrapolation, and inter-model robustness. Section 4 explores inter-scenario robustness, i.e., whether the patterns obtained for $CO_2$ are useful for pattern scaling other scenarios.

One common feature of all pattern scaling methods is that they are largely statistical approaches. While this is often suitable for obtaining scaling factors that accurately approximate simulations conducted with more complicated models (here, Earth

System Models), there are potential issues that are necessarily inherent to statistical approaches, primarily dealing with nonlinearity in the response and extrapolation. If the climate response is perfectly linear, then any pattern scaling method will work equally well and will be highly accurate. However, if the climate response is nonlinear, as might be expected to some degree, then any linear approximation will have reduced fidelity, and error will increase as one extrapolates to time periods farther from the training data set. Conversely, physically-based approaches are less prone to issues that arise from extrapolation, provided

that all of the relevant system dynamics are captured in the emulated pattern. There are many possibilities for physically-based approaches; here we evaluate one (described below) and compare it to the other two methods.

Through these investigations, we hope to better reveal which methods of pattern scaling of precipitation perform better than others. We will also provide some (limited) guidance as to which situations pattern scaling is likely to provide a computationally efficient, reasonably accurate result, versus which situations require actual simulation using Earth System Models.

## 15  2   Pattern Scaling Methods

### 2.1   Three Methods of Pattern Scaling for Precipitation

Pattern scaling involves approximating a timeseries of the pattern of change in a field of interest $\Delta B(\mathbf{x},t)$ by $\Delta \hat{B}(\mathbf{x},t)$:

$$\Delta B(\mathbf{x},t) \approx \Delta \hat{B}(\mathbf{x},t) = P(\mathbf{x})\Delta \bar{T}(t) \tag{1}$$

where $P(\mathbf{x})$ describes a time-invariant spatial pattern (the spatial dimension is denoted by $\mathbf{x}$), and $\Delta \bar{T}(t)$ describes a time-

20 varying (the time dimension is denoted by $t$) series of the change in global mean temperature, starting from a reference period $t = 0$ (often the preindustrial era). This notation will be used repeatedly throughout the manuscript. There are two commonly used methodologies for ascertaining $P(\mathbf{x})$: regression and epoch differencing (Barnes and Barnes, 2015). In the regression method, $P(\mathbf{x})$ is obtained by regressing $\Delta B(\mathbf{x},t) = B(\mathbf{x},t) - B(\mathbf{x},0)$ against $\Delta \bar{T}(t)$ at each point in $\mathbf{x}$. In the epoch method,

$$P(\mathbf{x}) = \frac{B(\mathbf{x},[k,n+k]) - B(\mathbf{x},[0,n])}{\bar{T}([k,n+k]) - \bar{T}([0,n])} \tag{2}$$

where the intervals $[0,n]$ and $[k,n+k]$ indicate averaging over $n$-year time periods at the beginning and end of the simulation, respectively. All values calculated are over a multi-model mean; Ruosteenoja et al. (2007) showed that pattern scaling for precipitation over a model mean outperforms results obtained from using single models. Frieler et al. (2012) argued that no single set of regression coefficients will be applicable to all grid points. We circumvent this issue by (for example) regressing $\Delta \bar{T}$ against $\Delta B$ at each grid point. By the results of Lynch et al. (2016), who showed excellent pattern scaling relationships for

temperature, this approach automatically accounts for correlations between local temperature and local precipitation changes.





In addition to these two methods, which were explored by Lynch et al. (2016) for use in pattern scaling temperature, we introduce a third, physically-based method. The physically-based reconstruction is founded on the work of Lau et al. (2013), who discovered a robust hydrological cycle response to global warming amongst the models participating in the Coupled Model Intercomparison Project Phase 5 (CMIP5; Taylor et al., 2012). The idea of this method is based on the "rich-get-richer, poor-get-poorer" concept wherein areas that receive high amounts of precipitation will receive more as global temperature increases, and areas with low precipitation will receive even less (e.g., Trenberth, 2011). Instead of finding differences in annual averages, this method calculates differences in statistics of precipitation intensity and spatial distribution.

The pattern $P(x)$ for the physically-based method is constructed as follows:

1. For each model, monthly mean precipitation values over the first 25 years of the simulation being evaluated are binned into low ($< 0.3$ mm day$^{-1}$), medium ($0.9 - 2.4$ mm day$^{-1}$), and high precipitation ($> 9$ mm day$^{-1}$). The same is done for a later epoch of length 25 years. The difference between the two epochs is then calculated, yielding three maps of precipitation change for low, medium, and high precipitation.

2. The multi-model mean of the results of Step 1 is calculated for each of the three maps. Map values are set to 0 where fewer than 65% of the models agree on the sign of the response. Then all three maps are added together.

3. For each epoch, global mean temperature is averaged over the entire 25-year period, and then the results for the two epochs are subtracted to obtain an estimate of change in global mean temperature over that period. The pattern $P(x)$ is then defined as the map from Step 2 divided by this temperature change.

Lau et al. (2013) found that for 14 models participating in CMIP5, there is a robust hydrological cycle response both in terms of frequency of occurrence of precipitation events, as well as the spatial distribution of precipitation change. Preliminary tests using a different set of models (Table 1 below, results not shown) indicate that we can replicate the findings and spatial patterns of precipitation change as depicted by Lau et al. (2013) rather well.

## 2.2 Methodology

In the following sections, we quantify differences between the reconstruction $\hat{B}$ and the actual model output $B$ via the root mean square (RMS) over the area-weighted difference $\hat{B} - B$, calculated as

$$\text{RMS} = \frac{\sqrt{\sum_x \left[\left(\hat{B}(x) - B(x)\right) \cdot A(x)\right]^2}}{\sqrt{\sum_x \left[A(x)\right]^2}} \qquad (3)$$

where $A(x)$ is the area of grid box $x$, and sums are calculated over all $x$.

Because Lau et al. (2013) only define their methodology for grid boxes between 60°S and 60°N, we compared RMS values restricted to that range with RMS values calculated over the entire globe. The results were quite similar in both cases (comparison not shown), so we only report RMS values calculated over the entire globe. This indicates that, on average, errors in the range of 60°S to 60°N are similar to errors at high latitudes, which is somewhat inconsistent with a conclusion of Tebaldi and



Arblaster (2014) that scalings that include zonal mean temperature have better fidelity to the actual model output because they can account for the effects of polar amplification. This indicates the potential for robustness of the physically-based method.

All of the analysis conducted here uses simulations from Earth System Models contributed to CMIP5. The models used in the bulk of the analysis in this study (Table 1, Group 1) are identical to those used by Lynch et al. (2016) with two exceptions (due to model output availability):

1. The present study used NorESM1-ME instead of NorESM1-M. NorESM1-ME includes prognostic biogeochemical cycling and has the capability of being emissions-driven, but when using concentration-driven scenarios (as is the case here), the two versions of the model will produce nearly identical results (Bentsen et al., 2013).

2. The present study used CMCC-CM instead of CMCC-CMS. The difference between these two versions is that CMCC-CMS has a fully-resolved stratosphere, whereas CMCC-CM is the lower-top version of the model (Davini et al., 2014; Sanna et al., 2013). Cagnazzo et al. (2013) describe some of the differences between these two models. In general, the models agree on qualitative climate features, although as might be expected, CMCC-CMS better matches observations in situations where a fully resolved stratosphere is important for capturing the effects, including dynamical feedbacks of stratospheric circulation and ozone chemistry on surface climate. Although these effects are non-negligible, they are generally of lower order than the changes that occur over the course of the scenarios analyzed in this study (to be discussed presently), so we anticipate that differences between these two models will not substantially affect results for the model mean.

Throughout this study, we evaluate three scenarios. The 1pctCO2 scenario involves a 1% per year increase in the $CO_2$ concentration, beginning at its preindustrial value. This simulation is run for 140 years to an approximate quadrupling of the $CO_2$ concentration. The RCP8.5 and RCP2.6 scenarios (Representative Concentration Pathways, or RCPs; Moss et al., 2010; Meinshausen et al., 2011) describe the results of two socioeconomic narratives that produce particular concentration profiles of greenhouse gases, aerosols, and other climatically relevant forcing agents over the 21st century. The RCP8.5 scenario reflects a "no policy" narrative, in which total anthropogenic forcing reaches approximately 8.5 W m$^{-2}$ in the year 2100. Conversely, the RCP2.6 scenario involves aggressive decarbonization, causing radiative forcing to peak at approximately 3 W m$^{-2}$ around 2050 and decline to approximately 2.6 W m$^{-2}$ at the end of the 21st century. Table 2 provides additional forcing details for the two RCP scenarios, as calculated by Hector (Hartin et al., 2015), a climate, carbon-cycle model that is used as the climate component of the Global Change Assessment Model (GCAM), a state-of-the-art Integrated Assessment Model. Both RCPs are appended to simulations of the historical period, for total simulation lengths of 251 years (1850–2100).

Throughout the remainder of the paper, subscripts on $P$, $\bar{T}$, $\hat{B}$, and $B$ are used to denote the scenario (e.g., RCP8.5), the model group (e.g., Group 2), or the years over which the patterns are computed (e.g., $1-50$). If there is no subscript specified, then the associated value corresponds to the Group 1 (see Table 1) multi-model mean of the 1pctCO2 simulation, averaged over years 116–140 of the simulation.

Statistical significance was calculated using Welch's $t$-test, which is analogous to a Student's $t$-test, but where the variances $s_1$ and $s_2$ of the two samples $x_1$ and $x_2$, respectively, do not need to be equal. We use this statistic here because the ensemble





for each method is small, and the ensemble pattern distribution is assumed to be normal. The test statistic is defined by

$$t = \frac{\bar{x}_1 - \bar{x}_2}{\sqrt{s_1^2/n_1 + s_2^2/n_2}} \tag{4}$$

where $n_1$ and $n_2$ are the number of models in each sample, respectively. Once the $t$ statistic is calculated for each grid box, the value in any given grid box is determined to be statistically significant if the test value exceeds a threshold computed from the inverse of the Student's $t$ cumulative probability distribution at the 97.5% confidence level (which is the 95% confidence level for a two-sample test). The number of degrees of freedom $df$ used to generate that threshold is approximated by the Welch-Satterthwaite Equation:

$$df = \frac{\left(s_1^2/n_1 + s_2^2/n_2\right)^2}{\frac{s_1^2/n_1}{n_1-1} + \frac{s_2^2/n_2}{n_2-1}} \tag{5}$$

In all figures, stippling is used to obscure values that are not statistically significant, i.e., the $t$-statistic failed to exceed the 95% confidence threshold.

## 3 Comparisons Between Pattern Scaling Methods for CO$_2$-Only Forcing

Figure 1 shows the baseline (preindustrial) precipitation pattern $B(x,0)$ and the scaling patterns $P(x)$ for each of the three pattern scaling methods generated from the Group 1 (see Table 1) model average for the 1pctCO2 simulation. The regression and epoch difference methods have very similar scaling patterns, and both are quite different from the comparatively more responsive physically-based method. All patterns show similar broad features: an increase in tropical precipitation with global warming, particularly over the oceans; increases at high latitudes, again over the oceans; and decreases in the South Pacific, North Atlantic, and South Indian Oceans, as well as Central America and the Mediterranean basin.

Figure 2 shows differences in $P(x)$ between each of the three methods. The epoch difference and regression methods have no differences greater in magnitude than 0.05 mm day$^{-1}$ K$^{-1}$, and no differences are statistically significant. In comparison to those two methods, the physically-based method is more responsive in the tropics (except the equatorial Pacific, where it is less responsive) and generally less responsive in other regions where the patterns show a nonzero response. Most of these differences in patterns are statistically significant.

### 3.1 Pattern Scaling for CO$_2$ Concentration Changes

Figure 3 shows a comparison between the actual model output (Group 1 averaged over the mean of years 116–140 of the 1pctCO2 simulation) and the three methods of reconstruction. All of the methods show qualitatively similar features. The physically-based reconstruction has too much precipitation in the tropics and Northern Hemisphere subtropics, especially over the tropical oceans. Tebaldi and Arblaster (2014) note that pattern scaling methodologies have difficulty in representing convection processes, so departures in these areas might be expected. The regression and epoch difference methods are quite similar to each other, and both reproduce the qualitative features of the actual model output well.





Figure 4 shows a more quantitative comparison between the different reconstruction methods and the actual model output. The physically-based method has larger error than the other two methods nearly everywhere. Overall error (RMS; Equation 3) is approximately 50 times greater than the other two methods (Table 3). Error in the regression and epoch different methods are very small (0.04 and 0.03 mm day$^{-1}$, respectively), and no region in the reconstruction is statistically different from the

actual model output. Conversely, the physically-based reconstruction often exceeds 25% error.

## 3.2   Interpolation/Extrapolation

In this section, we examine robustness of the methods to interpolation or extrapolation. If the scaling pattern $P(\mathbf{x})$ truly is time-invariant, then the results presented in this section will be identical to those previously discussed.

The poor performance of the physically-based method extends to extrapolation. Figure 5 shows the patterns $P(\mathbf{x})$ obtained

by conditioning the reconstructions only on years 1–50 of the 1pctCO2 simulation. In the physically-based and epoch difference methods, the second epoch is calculated over years 26–50 instead of years 116–140. In the regression method, the regression coefficients are calculated only using the first 50 years of simulation. The pattern $P(x)$ is nearly zero everywhere for the physically-based method, indicating that the statistics of precipitation do not differ appreciably between the two periods. This results in large areas of statistically significant differences from the patterns calculated using the full 140 years. Conversely, the

patterns calculated by using the regression and epoch difference methods only show small changes between the two periods, none of which is statistically significant.

Despite similarities, using patterns conditioned on the earlier period to reconstruct the precipitation in the later period (years 116–140) results in considerably poorer performance for the regression and epoch difference methods (Figure 6) than the results shown in Figure 4. RMS error for the regression and epoch difference methods increases by an order of magnitude

(Table 3), although few areas show statistically significant differences from the actual model output over this time period. RMS error for the physically-based method is reduced by a factor of three and is similar to the RMS error of the other two methods. The physically-based reconstruction still has areas of statistically significant error, but they are generally fewer and smaller in absolute magnitude.

Figure 7 shows results for interpolation, where the patterns are conditioned on the full 1pctCO2 simulation (years 116–

140), but the reconstruction predicts the average temperature in years 58-82 (halfway through the 1pctCO2 simulation). More specifically, $\hat{B} = P_{116-140}(\mathbf{x})\Delta\bar{T}(58-82)$. In general, the patterns for interpolation show similar qualitative features to those of reconstructing the later time period of years 116–140 (Figure 4). However, error is reduced by a factor of two for the physically-based method and increases by a factor of two for the regression and epoch difference methods. As before, no difference is statistically significant for the regression or epoch difference methods. The physically-based method has some

areas with statistically significant differences, but they are fewer and smaller in extent.

## 3.3   Inter-Model Robustness

In this section, we explore the role of the number of models in improving robustness of the prediction, as well as inter-model robustness of pattern scaling by comparing reconstructions with actual model output where the scaling pattern $P(\mathbf{x})$ is





conditioned on an entirely different set of models. More specifically, we examine two questions: (1) How does the prediction fidelity vary with the number of models used in the average? (2) If one conditions the pattern scaling on the average of Group 1, can one predict the response of Group 2 (or vice versa)?

Figure 8 shows the RMS error in the reconstruction (1pctCO2 simulation, averaged over years 116–140) as a function of
the number of models used in the comparison. This figure was created by randomly sampling the space of all 26 models listed in Table 1; each box/set of whiskers indicates 20 sets of random samples. Results ascertained from this figure parallel those discussed in previous sections: the regression and epoch difference methods have similar magnitudes of error (except for small numbers of models), and both have error that is approximately two orders of magnitude lower than the physically-based reconstruction. The values in Table 3 indicate that Group 1 (13 models) is not an outlier.

Both the physically-based reconstruction and regression methods show a dependence of RMS error on the number of models, whereas with the exception of low model numbers (<10), there is much lower dependence for the epoch method. However, except for low model numbers, none of the boxes/whiskers is substantially different from any of the others, leading us to conclude that each of the methods is largely robust to changes in the number of models used to carry out pattern scaling. Appendix 1 and the associated figures provide additional comparisons between the patterns generated for Groups 1 and 2.

## 3.4   Discussion of Pattern Scaling the Precipitation Response to $CO_2$

The regression and epoch difference methods show great promise in their usefulness as precipitation pattern scaling methods. Both are able to reconstruct the changes in precipitation due to $CO_2$ increases with errors of less than 5% in every region of the globe (Figure 4). Conversely, the physically-based method has comparatively poor performance, with error regularly exceeding 25%. This method often scales too strongly with global mean temperature change. The test of extrapolation shows that for the
time periods analyzed here, using $P(x) = 0$ would result in better absolute performance than using the physically-based pattern $P(x)$ depicted in Figure 2.

However, the physically-based method shows robustness in several ways that the regression and epoch difference methods do not. One example is with interpolation, where error drops substantially for the physically-based method but increases for the other two. Another is related to inter-model robustness: the pattern of error is relatively unchanged for the physically-based
method, regardless of the group of models that is used (Appendix 1), whereas the pattern shows increased error in many places for the other methods. The overall performance of the physically-based method is still worse in all cases, but these results suggest that if the overall bias in the physically-based method could be reduced or corrected, it holds great promise in being a useful pattern scaling method, particularly in situations (like the ones explored in Sections 3.2 and 3.3) where statistically-based or other non-physical methods might be expected to perform less well.

Like the temperature pattern scaling results of Lynch et al. (2016), we find that the regression and epoch difference methods have similar performance. In the present work, we find that the epoch difference method slightly outperforms the regression method, but the differences are relatively minor. Given the slight advantages in computational expense and reduced data input requirements, we profess a slight preference for using the epoch difference method to generate scaling patterns for the precipi-





tation response to $CO_2$-induced global warming. In the next section, we explore a more broad application of pattern scaling by including non-$CO_2$ forcings.

## 4  Pattern Scaling for Non-$CO_2$ Forcings

In this section, we compare the patterns and reconstructions between scenarios, primarily related to the RCP8.5 and 1pctCO2
simulations. We do this first as a test of robustness: does any one of the three methods perform "better" for $CO_2$-only sim-
ulations versus RCP8.5? If the fidelity of the reconstruction to the actual model output is similar for the two scenarios, then
subtracting the reconstructions conditioned on RCP8.5 and 1pctCO2 could reveal a scaling pattern for non-$CO_2$ forcing. We
note that this is one of the few ways of ascertaining the non-$CO_2$ response pattern without running separate simulations both
with and without $CO_2$ forcing—without a scaling method to normalize for similar climate conditions, there is no way of
obtaining meaningful results from directly subtracting a 1pctCO2 simulation from an RCP8.5 simulation. (The approach dis-
cussed here is analogous to the methodology of Herger et al. (2015), but where they attempted to ascertain similarities between
patterns for a given change in global mean temperature, we are interested in the differences.)

We note several caveats with this approach. One is that, based on the results of Herger et al. (2015), the reconstructions of
RCP8.5 and 1pctCO2 are likely to be similar for a given temperature change because the dominant forcing in RCP8.5 is $CO_2$
(see Table 2). As such, ascertaining the non-$CO_2$ signal could be limited by low signal-to-noise ratios. A second caveat, one
more germane to pattern scaling, is to ascertain whether the non-$CO_2$ pattern obtained from RCP8.5 can be used to reconstruct
the non-$CO_2$ precipitation change for a different scenario. There is no *a priori* reason to expect this will work, as different
scenarios have different combinations of forcings, but as long as the $CO_2$ portion dominates the response, such endeavors may
still be useful. In Section 4.3, we investigate this problem using an extreme case, where we ascertain the scaling patterns from
an RCP8.5 simulation and use them to attempt to reconstruct the RCP2.6 simulation.

### 4.1  Inter-Scenario Differences

Figure 9 shows the RCP8.5 scaling pattern $P_{\text{RCP8.5}}(\mathbf{x})$ and the difference from the $CO_2$-only pattern. Patterns are nearly
identical to those in Figure 1. The physically-based pattern shows some statistically significant changes, particularly in the
tropics and at high latitudes. The regression and epoch difference methods show no differences exceeding 0.1 mm day$^{-1}$
K$^{-1}$ in magnitude and no statistically significant differences of any magnitude. This figure reinforces the findings of Herger
et al. (2015) that patterns generated from commonly used scaling methods (regression and epoch difference) do not differ
appreciably between scenarios, so pattern scaling can be accomplished by using periods in different scenarios with the same
global mean temperature change.

Figures 10 and 11 show this in practice, where the reconstruction $\hat{B}$ is built on the RCP8.5 pattern, multiplied by $\Delta\bar{T}$
averaged over years 227–251 and 116–140, respectively. The physically-based method qualitatively and quantitatively matches
the results in Figure 4. For the regression and epoch difference methods, the reconstructed precipitation response in Figure 10
is generally too strong in the tropics and too weak in the midlatitudes (which is the same pattern in Figure 4), but Figure 11





shows the opposite pattern. None of these differences is statistically significant, and the RMS error is approximately the same in both figures (0.09–0.10 mm day$^{-1}$ K$^{-1}$; 2–3 times greater than the error in Figure 4), but they suggest that there is a distinct non-$CO_2$ pattern that, while small, is still important in explaining precipitation differences in periods with large temperature change.

Figure 12 provides descriptions of the actual precipitation effects of both $CO_2$ and non-$CO_2$ forcing. (In the following, we omit discussion of the results from the physically-based method due to the aforementioned strong biases in the results.) Large features of the non-$CO_2$ response are the opposite of the $CO_2$ response, indicating an offsetting. In the non-$CO_2$ forcing case, equatorial precipitation is weakened, possibly indicating a Southward shift in the intertropical convergence zone. Precipitation over East Asia and the maritime continent is also weaker, while precipitation in Amazonia, the South Pacific, the North Atlantic,

and the Northern Hemisphere subtropical Pacific is stronger. It would be a useful future area of investigation to ascertain whether these patterns of precipitation change arise in climate models forced with appropriate non-$CO_2$ forcings, and if so, the mechanistic reasons why these responses are different from $CO_2$ forcing. Such investigations would also aid in understanding the robustness of these signals, i.e., what portion of the reported response in Figure 12 is signal versus noise. The results in Figure 12 also reinforce the conclusions of Frieler et al. (2012), who argue that the scaling patterns from one scenario

are not translatable to scaling patterns for another scenario if the two scenarios are driven by different forcing. Even though Figure 9 shows that the patterns $P_{\mathrm{RCP8.5}}$ and $P_{\mathrm{1pctCO2}}$ are nearly identical, even small differences can affect reconstructions of precipitation change for large values of $\Delta\bar{T}$.

## 4.2  Non-$CO_2$ Forcing Pattern

Here we calculate a non-$CO_2$ pattern for use in pattern scaling. We begin by assuming that the effects of $CO_2$ forcing and

non-$CO_2$ forcing are separable, that is, that there are no nonlinear interactions between the two forcings that would produce a non-additive response. Although this assumption is not strictly true, it is approximately true to a sufficient degree that such calculations are useful (MacMartin et al., 2015; MacMartin and Kravitz, 2016). Following the notation in Equation 1, separability means that

$$\Delta\hat{B}_{\mathrm{RCP8.5}} = \Delta\bar{T}_{\mathrm{CO_2}} P_{\mathrm{CO_2}} + \Delta\bar{T}_{\mathrm{non-CO_2}} P_{\mathrm{non-CO_2}} \qquad (6)$$

We set $P_{\mathrm{CO_2}}$ equal to $P_{\mathrm{1pctCO2}}$ (from Section 3), because if pattern scaling holds, the time-invariant pattern of $CO_2$ forcing should be identical, regardless of the scenario from which it is derived. $P_{\mathrm{non-CO_2}}$ is assumed to be $P_{\mathrm{RCP8.5}} - P_{\mathrm{CO_2}}$. This is the inherent assumption of the linear pattern scaling approach. If the approach fails, it is because this pattern does not represent actual non-$CO_2$ forcing. To calculate $\Delta\bar{T}_{\mathrm{CO_2}}$, we assume that global mean temperature scales linearly with radiative forcing (e.g., Gregory et al., 2004), and radiative forcing is known to scale logarithmically with the $CO_2$ concentration (Myhre et al.,

1998). Performing linear regression of $\log_2([CO_2])$ against global mean temperature change in the 1pctCO2 simulation yields a slope of $\alpha = 2.40$, an intercept of $\beta = -19.89$, and an $R^2$ value of 0.99. (Brackets $[\cdot]$ indicate the $CO_2$ concentration.) Then $\bar{T}_{\mathrm{CO_2}} = \alpha\log_2([CO_2]) + \beta$. $\bar{T}_{\mathrm{non-CO_2}}$ is calculated as the residual $\bar{T}_{\mathrm{RCP8.5}} - \bar{T}_{\mathrm{CO_2}}$.





Figure 13 shows all of the aforementioned $\bar{T}$ values, plotted as a function of the $CO_2$ concentration. The temperature contribution of the non-$CO_2$ part increases with the $CO_2$ concentration, which monotonically increases with time in the RCP8.5 simulation. This is consistent with the design of the RCP8.5 scenario, in which non-$CO_2$ radiative forcing increases over the period 2000–2100 (Table 2), largely due to a doubling of the methane concentration over this period. This change in forcing

corresponds to a non-$CO_2$ induced temperature change (green line in Figure 13) from 0.31 to 1.36 K.

We next explore the ability of this decomposition to reconstruct actual model output for the end of the RCP8.5 scenario (Figure 14) and a historical period in the late 20th century (Figure 15). Qualitatively, the results for the physically-based method are similar to those of Figure 10 (RMS error is approximately 44% larger), whereas the differences between the reconstruction and the actual model output ($\hat{B} - B$) have the opposite sign as compared to Figure 10 for the other two methods (RMS error is

slightly higher but of a similar magnitude). Conversely, Figure 15 shows greater similarity to Figure 10 for the epoch difference and regression methods (RMS error is nearly identical); while the physically-based method shows similar qualitative features in both Figures 10 and 15, the magnitude of error is reduced in Figure 15 by nearly a factor of 10. The epoch difference and regression methods show a hemispheric contrast in error in the historical period, but in the later, future period, the errors have more similar patterns to those in previously described figures.

### 4.3   Scaling to Predict Other Scenarios

The final stage of inter-model exploration is to see how well the $CO_2$ and non-$CO_2$ patterns generated from one scenario can be used on another scenario. Here we choose the extreme case of predicting the pattern of precipitation change in RCP2.6, based on the patterns calculated from RCP8.5. In this scenario, the $CO_2$ concentration peaks and then drops slightly (Table 2). The non-$CO_2$ forcing comprises 29% of the total forcing in RCP8.5 in 2100 and 32% of the total forcing in RCP2.6 in 2100,

according to simulations using Hector (Hartin et al., 2015).

Figure 16 shows the effectiveness of this reconstruction process. The physically-based method shows a similar pattern of error, but with reduced magnitude. The epoch difference and regression methods show too much $CO_2$ response and not enough non-$CO_2$ response as indicated by the patterns displayed in Figure 15. This can potentially be explained by the forcing values given in Table 2. Although the non-$CO_2$ forcing is approximately the same percentage of the total forcing in both the RCP8.5

and RCP2.6 simulations, the absolute forcing values are such that only 0.37 W m$^{-2}$ of non-$CO_2$ forcing is exerted in RCP2.6 in the year 2100. This indicates that global mean temperature change in RCP2.6 due to non-$CO_2$ forcing ($\Delta\bar{T}_{\mathrm{RCP2.6,non-CO_2}}$) is small, and as such, the values depicted in Figure 16 are almost entirely due to $CO_2$ forcing. Although more formal study is needed, this indicates that the non-$CO_2$ forcing in RCP2.6 is insufficiently large to overcome issues with low signal-to-noise ratios in reconstructing patterns of precipitation change using this sort of decomposition.

One notable feature in Figure 16 is a difference at high latitudes due to polar amplification that is more pronounced in RCP8.5 than RCP2.6. This is an example of a nonlinearity that cannot be well captured by linear pattern scaling; future work involving scaling by zonal mean temperature (as was suggested by Tebaldi and Arblaster, 2014) could show promise in improving these sorts of inter-scenario comparisons.





## 4.4 Discussion of Pattern Scaling for Non-CO$_2$ Forcings

In general, the pattern scaling results depicted in Section 4 are consistent with previous studies. Herger et al. (2015) found that the patterns between scenarios are rather similar, which Figure 9 confirms. However, the results for pattern scaling may be scenario-dependent (Figure 12) if global mean temperature change ($\Delta \bar{T}$) is sufficiently large, which confirms the conclusions of Frieler et al. (2012).

The patterns of change indicate some degree of nonlinearity in the response, particularly in that Figures 10 and 11 show opposite signs of error. Because the relationships between CO$_2$ concentration and precipitation are quite linear (Section 3), we conclude that the nonlinearities are due to the non-CO$_2$ response. Without conducting pattern scaling analyses on single forcing runs, we are unable to ascertain the exact sources of nonlinearity. However, this does explain in part why the RCP8.5 pattern was unable to reproduce some of the features of the RCP2.6 response, in that they have substantially different magnitudes of non-CO$_2$ forcing, likely due to different combinations of forcing agents. We note that the differences between RCP8.5 and RCP2.6 are as extreme as was available, and the decomposition into CO$_2$ and non-CO$_2$ components may be more effective for scenarios that are more similar, particularly in their non-CO$_2$ components.

## 5 Conclusions

We have explored three different methods of pattern scaling for precipitation, with a focus on robustness to interpolation/extrapolation, inter-model variations, and inter-scenario differences. The physically-based method has substantially worse performance than the other two methods but shows some features of robustness that could be advantageous if overall biases in the method could be reduced. The regression and epoch difference methods have excellent, approximately similar performance.

One of the features that emerged was that the CO$_2$ and non-CO$_2$ components offset each other. Without a detailed assessment of the different single forcing agents that comprise the non-CO$_2$ patterns, we are unable to provide a mechanistic understanding of the causes of these features, but we highlight this as a promising area of future research.

Most of the errors that arise for any of the methods are either in areas dominated by convection (predominantly over the tropical oceans) or at high latitudes. Both of these areas are large sources of nonlinear responses to global mean temperature change, so pattern scaling might not be expected to perform well in these areas. The physically-based method does include a bin for high precipitation, so despite providing too much precipitation in these regions, it is the best equipped to deal with these sources of nonlinearity. Moreover, the approach of Tebaldi and Arblaster (2014) of using zonal mean temperature as a scaling parameter may prove useful in accounting for errors at high latitudes.

In terms of the usefulness of pattern scaling of precipitation, because the regression and epoch difference methods perform well over most land regions, they are likely quite suitable for a variety of applications, including societal models (like Integrated Assessment Models or impacts models) that mostly deal with land areas. If one's application requires good performance over tropical oceans, then pattern scaling may no longer be appropriate, and instead output from the full Earth System Model may be required. However, given the difficulties many Earth System Models have with proper representations of convective processes





and the resulting precipitation biases those difficulties cause (e.g., Song and Zhang, 2009), there may be doubts as to how well Earth System Models represent precipitation in these areas in the first place.

The results presented here have applications that extend beyond providing libraries of scaling patterns for Integrated Assessment Models. Another more speculative application involves efficacy of climate forcings. Kravitz et al. (2015) developed

a method of comparing forcing agents via analyses of their rapid adjustments (fast responses), that is, their responses in the absence of global mean temperature change. If our method of decomposing the response into $CO_2$ and non-$CO_2$ components could be extended to single forcings, then one could isolate the feedback responses (slow responses), which are the portions of the responses that depend on global mean temperature change. Thus, there is potential to provide a more quantitative intercomparison of the different effects of climate forcing agents.

**6   Code and/or Data Availability**

All computations were performed using MATLAB 2012b, developed by MathWorks. The libraries of patterns for each method will be included in an upcoming official code release of the Global Change Assessment Model (GCAM). All code used to generate figures in this manuscript will be publicly released through a code repository.

*Author contributions.*   All authors collectively designed the study. BK was the lead on conducting analysis and writing the paper, with input

from the other authors at all stages.

*Acknowledgements.*   This research is based on work supported by the US Department of Energy, Office of Science, Integrated Assessment Research Program. We acknowledge the World Climate Research Programme's Working Group on Coupled Modelling, which is responsible for CMIP, and we thank the climate modeling groups for producing and making available their model output. For CMIP the U.S. Department of Energy's Program for Climate Model Diagnosis and Intercomparison provides coordinating support and led development of software

infrastructure in partnership with the Global Organization for Earth System Science Portals. The Pacific Northwest National Laboratory is operated for the U.S. Department of Energy by Battelle Memorial Institute under contract DE-AC05-76RL01830.



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



**Table 1.** Models used in the present analysis. Most of the analysis was conducted using the models in Group 1. Additional investigations described in Section 3.3 (inter-model robustness) were conducted using the models in both Group 1 and Group 2. All model names listed are the standard names used in contributions to the Coupled Model Intercomparison Project Phase 5 (CMIP5; Taylor et al., 2012). Knutti et al. (2013) provide an excellent description of these models and their provenance.

| Group 1 | Group 2 |
|---|---|
| ACCESS1.0 | ACCESS1.3 |
| CanESM2 | BCC-CSM1.1 |
| CCSM4 | BCC-CSM1.1M |
| CMCC-CM | BNU-ESM |
| CNRM-CM5 | CESM1-BGC |
| CSIRO-Mk3.6 | CESM1-CAM5 |
| GFDL-CM3 | FGOALS-g2 |
| HadGEM2-ES | GFDL-ESM2M |
| INMCM4 | IPSL-CM5A-LR |
| IPSL-CM5A-MR | IPSL-CM5B-LR |
| MIROC-ESM | MIROC5 |
| MPI-ESM-MR | MPI-ESM-LR |
| NorESM1-ME | MRI-CGCM3 |

**Table 2.** Radiative forcing values (W m$^{-2}$) for RCP8.5 and RCP2.6 in 2000, 2050, and 2100. $CO_2$ forcing and total forcing were calculated using the simple climate model Hector (Hartin et al., 2015). Non-$CO_2$ forcing is calculated as the difference between total and $CO_2$ forcing. Percentages in parentheses indicate the percentage of the total forcing.

| | 2000 | 2050 | 2010 |
|---|---|---|---|
| $CO_2$ forcing (RCP8.5) | 1.226 | 3.289 | 6.167 |
| Total forcing (RCP8.5) | 1.991 | 5.049 | 8.686 |
| non-$CO_2$ forcing (RCP8.5) | 0.765 (38%) | 1.760 (35%) | 2.519 (29%) |
| $CO_2$ forcing (RCP2.6) | 1.267 | 2.174 | 1.765 |
| Total forcing (RCP2.6) | 2.066 | 3.195 | 2.601 |
| non-$CO_2$ forcing (RCP2.6) | 0.799 (39%) | 1.021 (32%) | 0.836 (32%) |





**Table 3.** RMS error values calculated over the entire globe (Equation 3) for each of the figures. All units are in mm day$^{-1}$ for differences (Figures 4, 6, 7, 10, 11, 14, 15, 16, 18, and 19) or mm day$^{-1}$ K$^{-1}$ for patterns (Figures 5, 9, and 17).

|  | Physically-based | Regression | Epoch difference |
|---|---|---|---|
| Figure 4 | 1.74 | 0.04 | 0.03 |
| Figure 5 | 0.42 | 0.07 | 0.08 |
| Figure 6 | 0.55 | 0.27 | 0.33 |
| Figure 7 | 0.81 | 0.07 | 0.07 |
| Figure 9 | 0.42 | 0.02 | 0.02 |
| Figure 10 | 1.88 | 0.10 | 0.10 |
| Figure 11 | 2.90 | 0.09 | 0.09 |
| Figure 14 | 2.71 | 0.16 | 0.14 |
| Figure 15 | 0.22 | 0.09 | 0.09 |
| Figure 16 | 1.10 | 0.21 | 0.20 |
| Figure 17 | 0.30 | 0.05 | 0.04 |
| Figure 18 | 1.69 | 0.10 | 0.09 |
| Figure 19 | 1.53 | 0.04 | 0.03 |





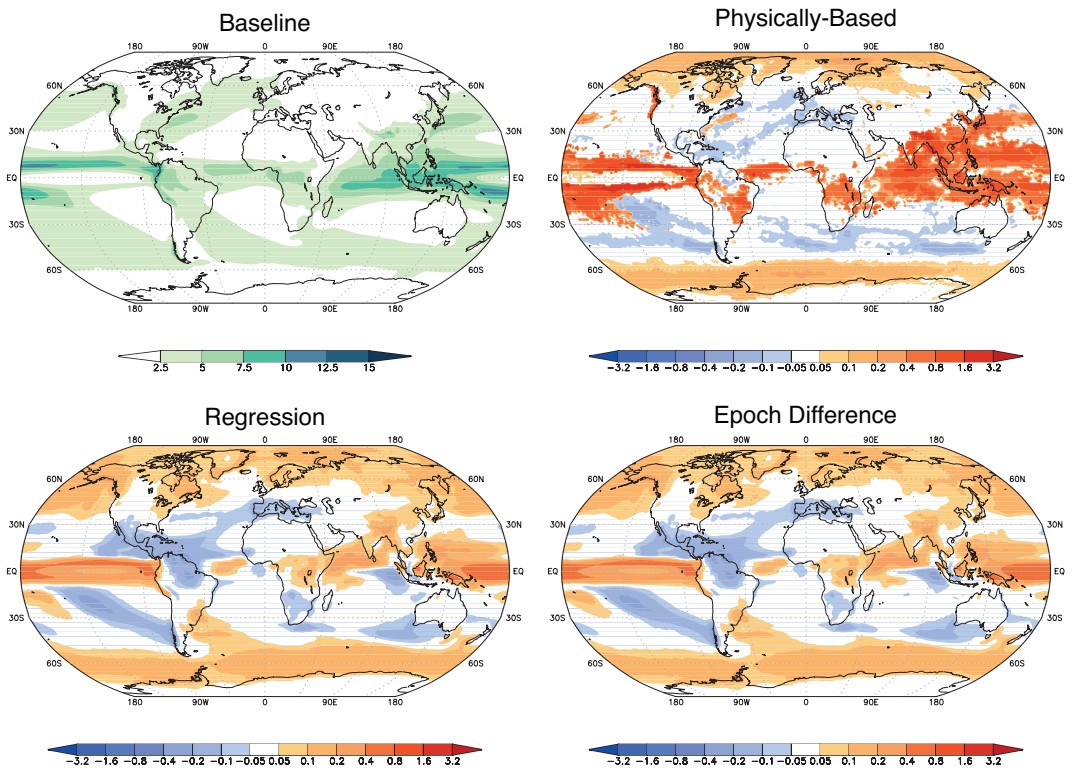

**Figure 1.** The components needed for pattern scaling of the precipitation response to $CO_2$ forcing, averaged over the 13 models in Group 1 (Table 1). Top-left shows the baseline precipitation pattern for the multi-model average: $B(x,0)$ in Equation 1 (mm day$^{-1}$; averaged over years 1–25 of the 1pctCO2 simulation). All other panels show the time-invariant pattern $P(\mathbf{x})$ in Equation 1 (mm day$^{-1}$ K$^{-1}$) for the physically-based method (top-right), the regression method (bottom-left), and the epoch difference method (bottom-right).



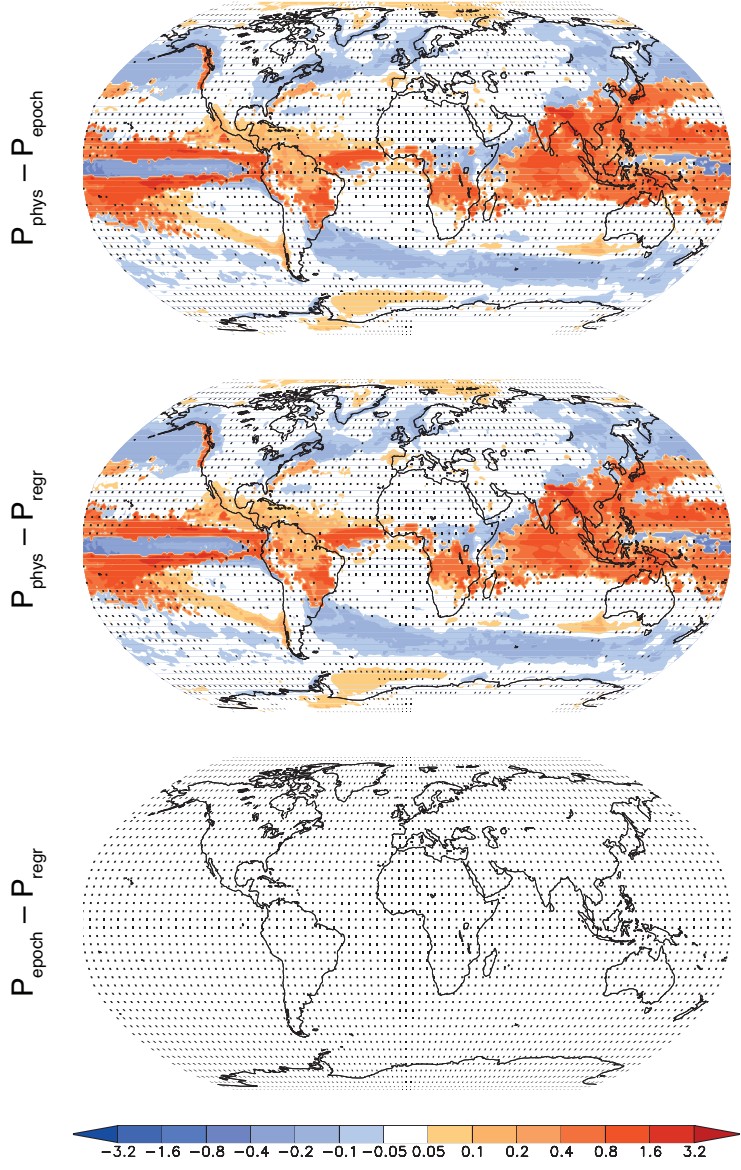

**Figure 2.** Differences in the precipitation scaling pattern $P(\mathbf{x})$ (Equation 1). Top panel shows $P_{\text{phys}} - P_{\text{epoch}}$, middle panel shows $P_{\text{phys}} - P_{\text{regr}}$, and bottom panel shows $P_{\text{epoch}} - P_{\text{regr}}$, where *phys* denotes the physically-based method, *regr* denotes the regression method, and *epoch* denotes the epoch difference method. All units are mm day$^{-1}$ K$^{-1}$ and were calculated for the multi-model average of Group 1 (Table 1). Stippling indicates a lack of statistical significance in the pattern of differences (Section 2.2).





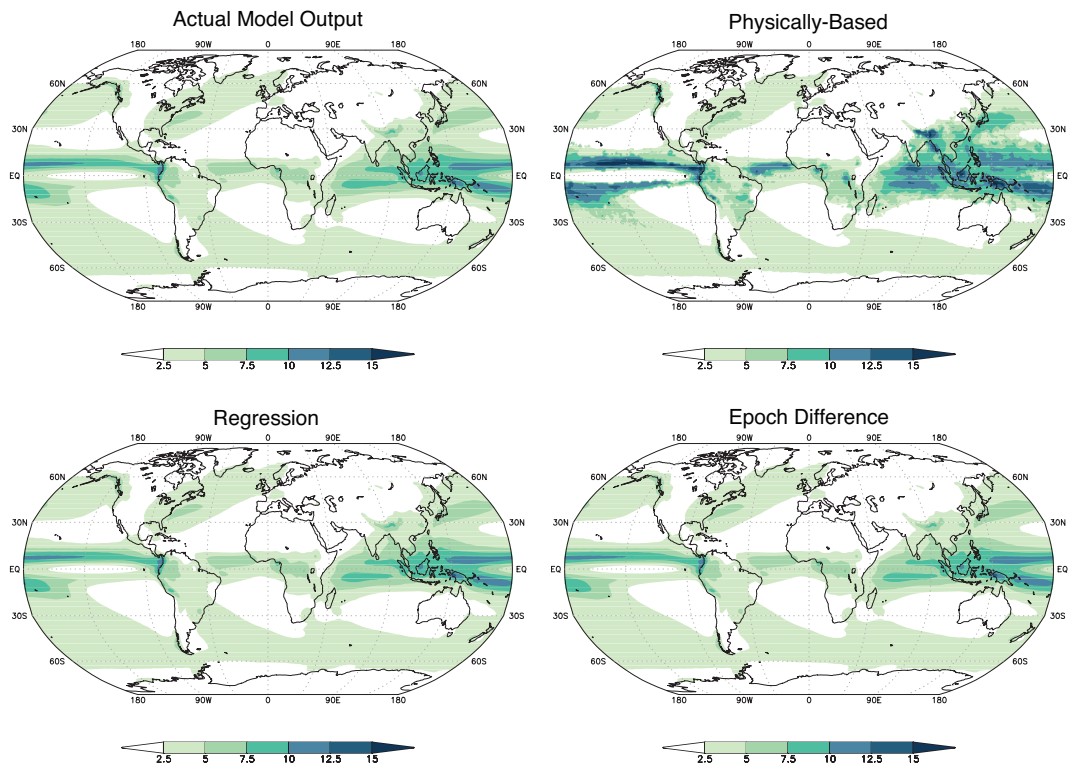

**Figure 3.** Comparison between the actual Group 1 multi-model average precipitation output (top-left) and the reconstructions produced by pattern scaling ($\hat{T}$ in Equation 1). All values are in mm day$^{-1}$ and represent averages over years 116–140 of the 1pctCO2 simulation. Top-right shows the physically-based method, bottom-left shows the regression method, and bottom-right shows the epoch difference method.





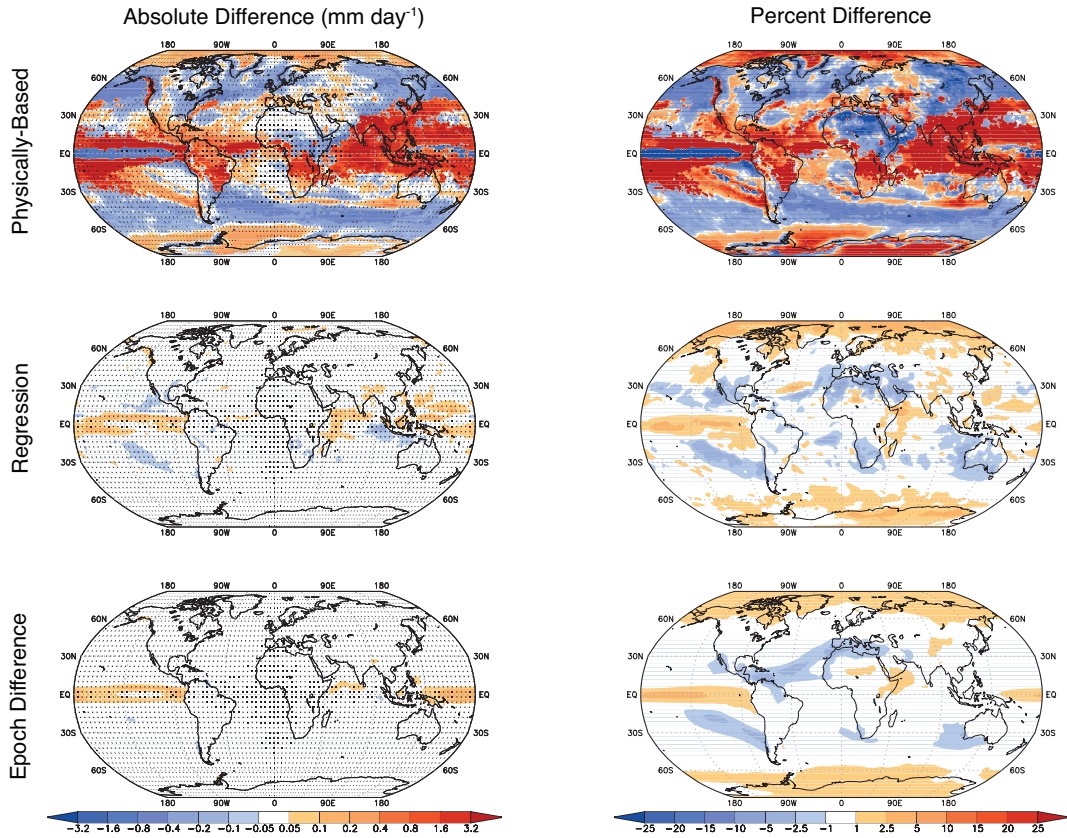

**Figure 4.** Differences between the reconstructions produced by pattern scaling ($\hat{B}$) and the actual model output for precipitation ($B$). Left column shows absolute values of $\hat{B} - B$ (mm day$^{-1}$), and right column shows percent change. Top row shows results for the physically-based method, middle row shows the regression method, and bottom row shows the epoch difference method. All values are calculated for a Group 1 multi-model average for the 1pctCO2 simulation over the years 116-140. Stippling indicates a lack of statistical significance in the pattern of differences (Section 2.2).





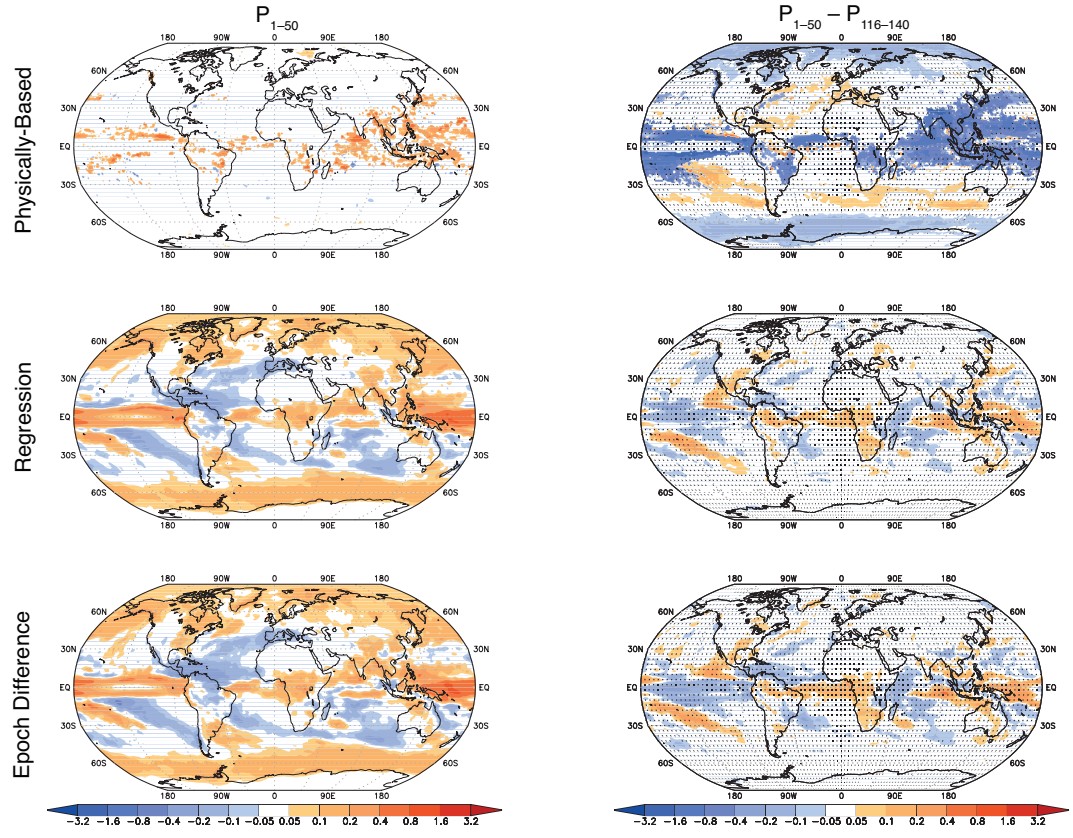

**Figure 5.** Differences in the precipitation scaling pattern $P(\mathbf{x})$ (Equation 1) when different time periods are used to construct the pattern (years 1–50 versus years 116–140 of the 1pctCO2 simulation). Left column shows values of $P_{1-50}$, and right column shows values of $P_{1-50} - P_{116-140}$ (mm day$^{-1}$ K$^{-1}$). Values in subscripts denote that the associated quantities are calculated from an average over those years. Top row shows results for the physically-based method, middle row shows the regression method, and bottom row shows the epoch difference method. All values are calculated for a Group 1 multi-model average for the 1pctCO2 simulation. Stippling indicates a lack of statistical significance in the pattern of differences (Section 2.2).





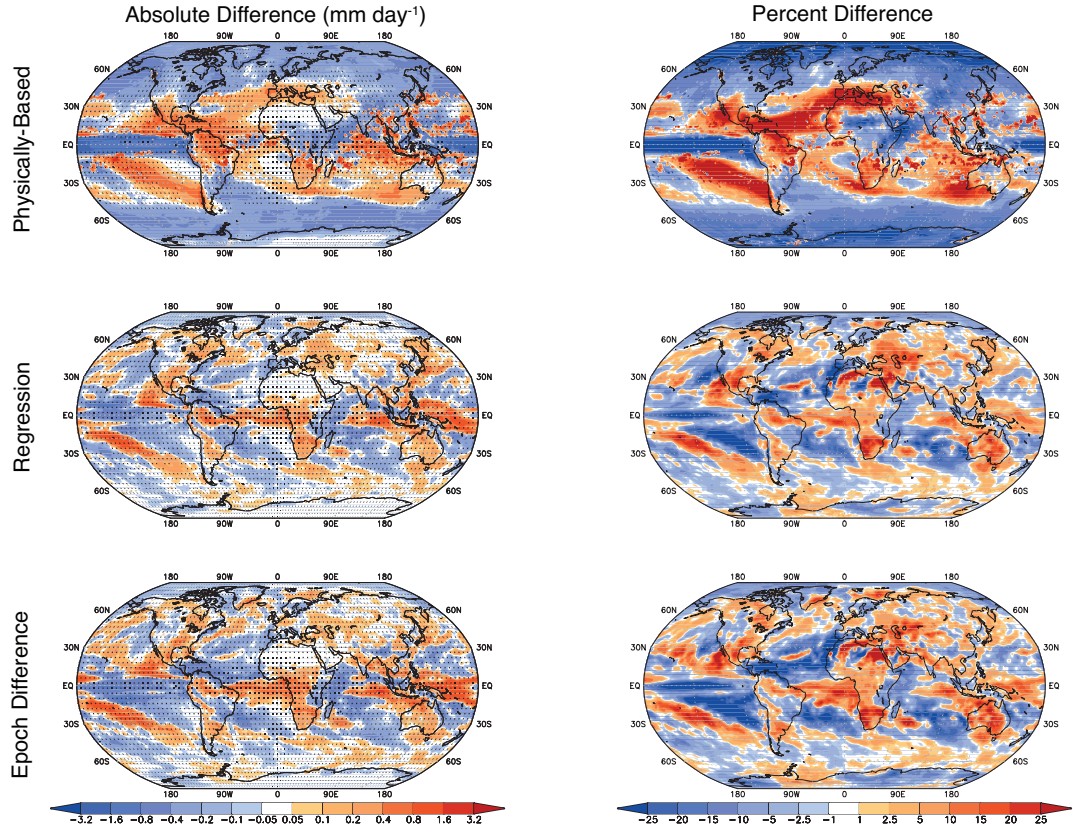

**Figure 6.** As in Figure 4 but where the reconstruction $\hat{B}$ is built on the pattern $P$ for years 1–50 (Group 1 average of the 1pctCO2 simulation), and global mean temperature $\Delta \bar{T}$ is averaged over years 116–140. That is, $\hat{B} = P_{1-50}(\mathbf{x})\Delta \bar{T}(116 - 140)$. Results shown are for the difference between the reconstruction and the actual model output $\hat{B} - B(\mathbf{x}, 116 - 140)$.





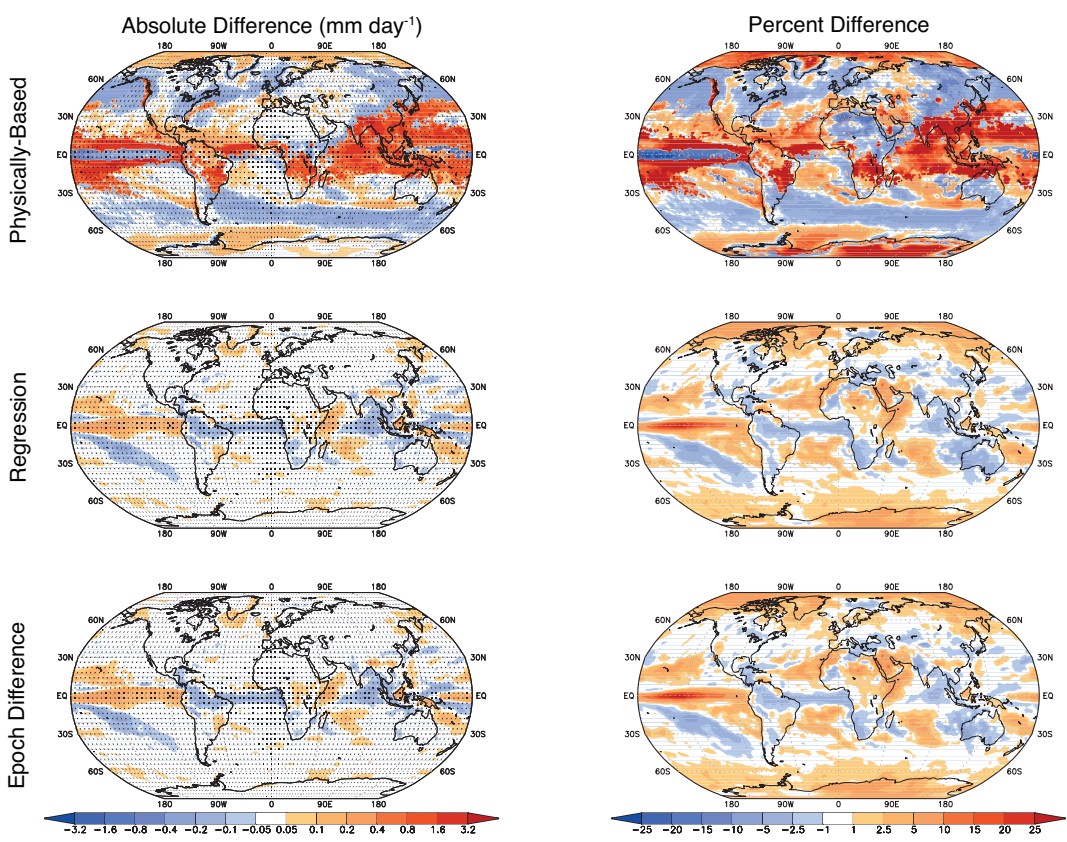

**Figure 7.** As in Figure 4 but where the reconstruction $\hat{B}$ is built on the pattern $P$ for years 116–140 (Group 1 average of the 1pctCO2 simulation), and global mean temperature $\Delta\bar{T}$ is averaged over years 58–82. That is, $\hat{B} = P_{116-140}(\mathbf{x})\Delta\bar{T}(58-82)$. Results shown are for the difference between the reconstruction and the actual model output $\hat{B} - B(\mathbf{x}, 58 - 82)$.





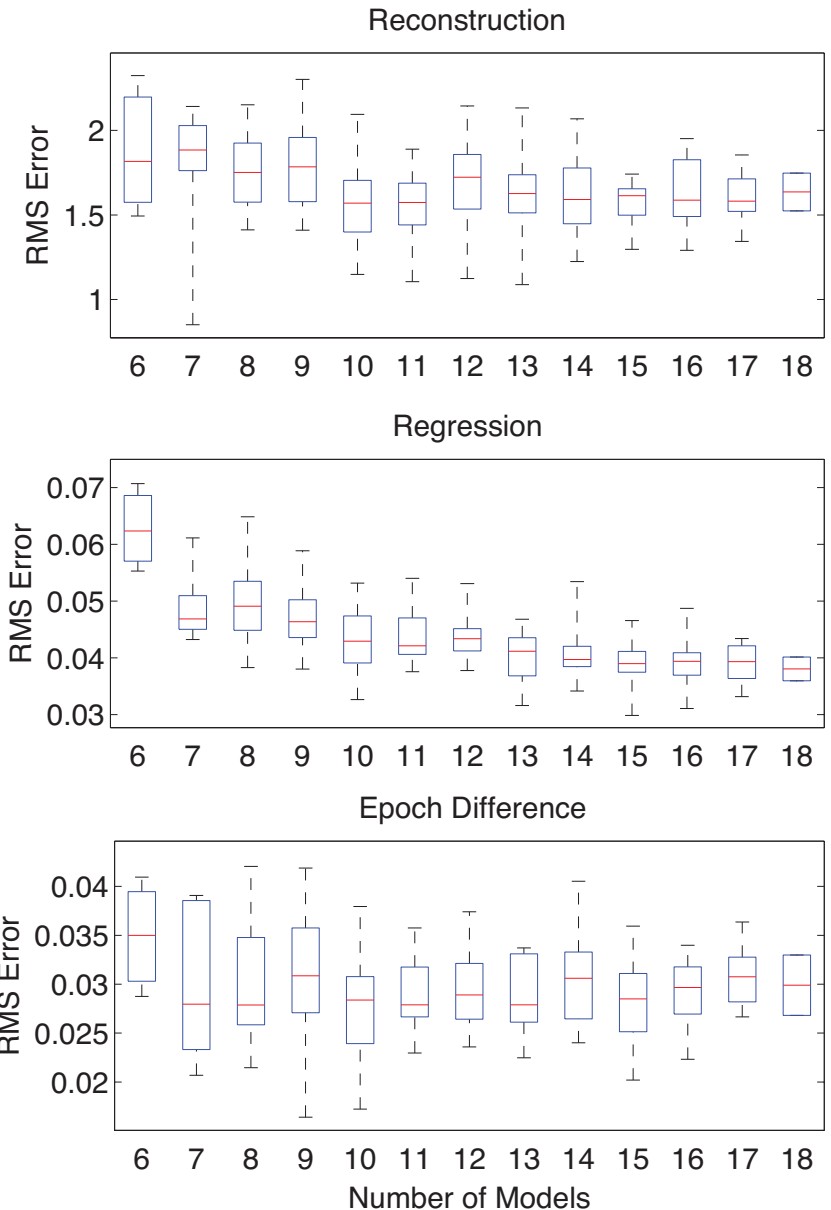

**Figure 8.** Root mean square (RMS) error (Equation 3, calculated on the difference between the reconstruction and actual model output $\hat{B} - B$) as a function of the number of models used to conduct the scaled precipitation reconstruction. Models were chosen randomly from a set of 26 models (Table 1). All values are calculated over an average of years 116–140. Each box in the plots represents 20 sets of models: red lines indicate median values, blue boxes indicate the 25th and 75th percentiles, and whiskers indicate the full range of model response among the 20 sets of models.





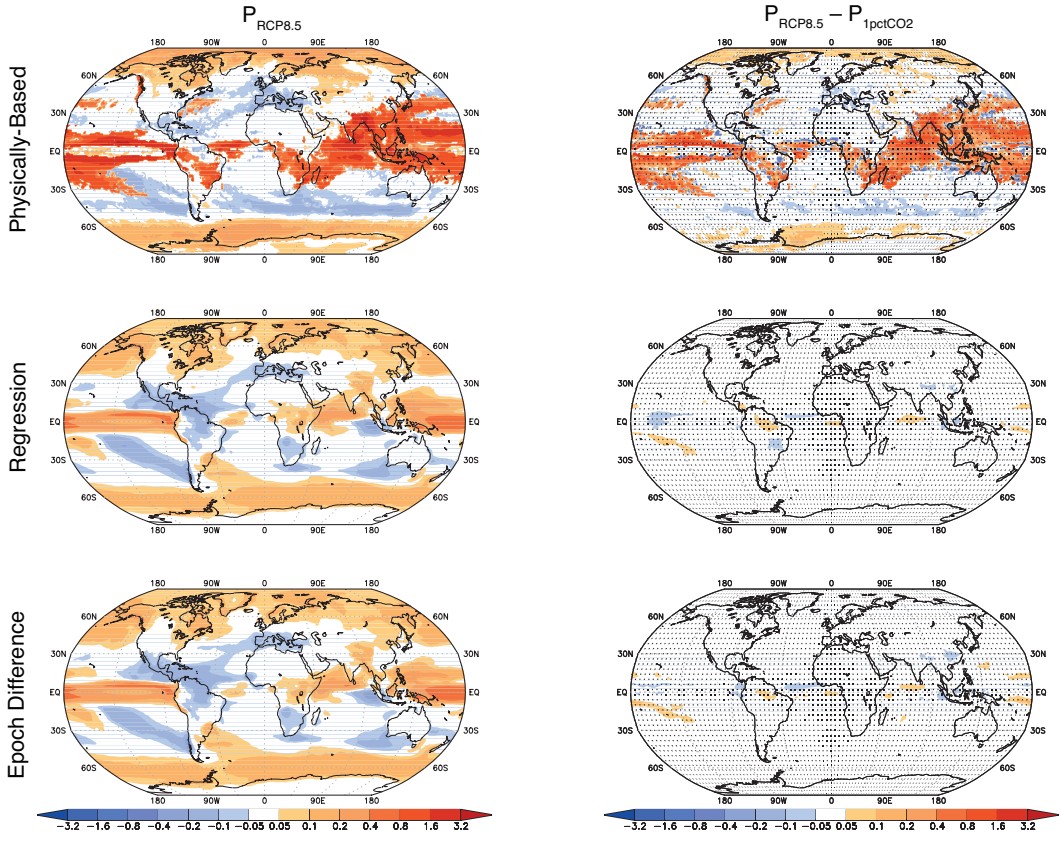

**Figure 9.** Differences in the precipitation scaling pattern $P(\mathbf{x})$ (Equation 1) when different scenarios are used to construct the pattern (RCP8.5 vs 1pctCO2). Left column shows values of $P_{\mathrm{RCP8.5}}$, and right column shows values of $P_{\mathrm{RCP8.5}} - P_{\mathrm{1pctCO2}}$ (mm day$^{-1}$ K$^{-1}$). Top row shows results for the physically-based method, middle row shows the regression method, and bottom row shows the epoch difference method. All values are calculated for a Group 1 multi-model average for the 1pctCO2 simulation. Stippling indicates a lack of statistical significance in the pattern of differences (Section 2.2).





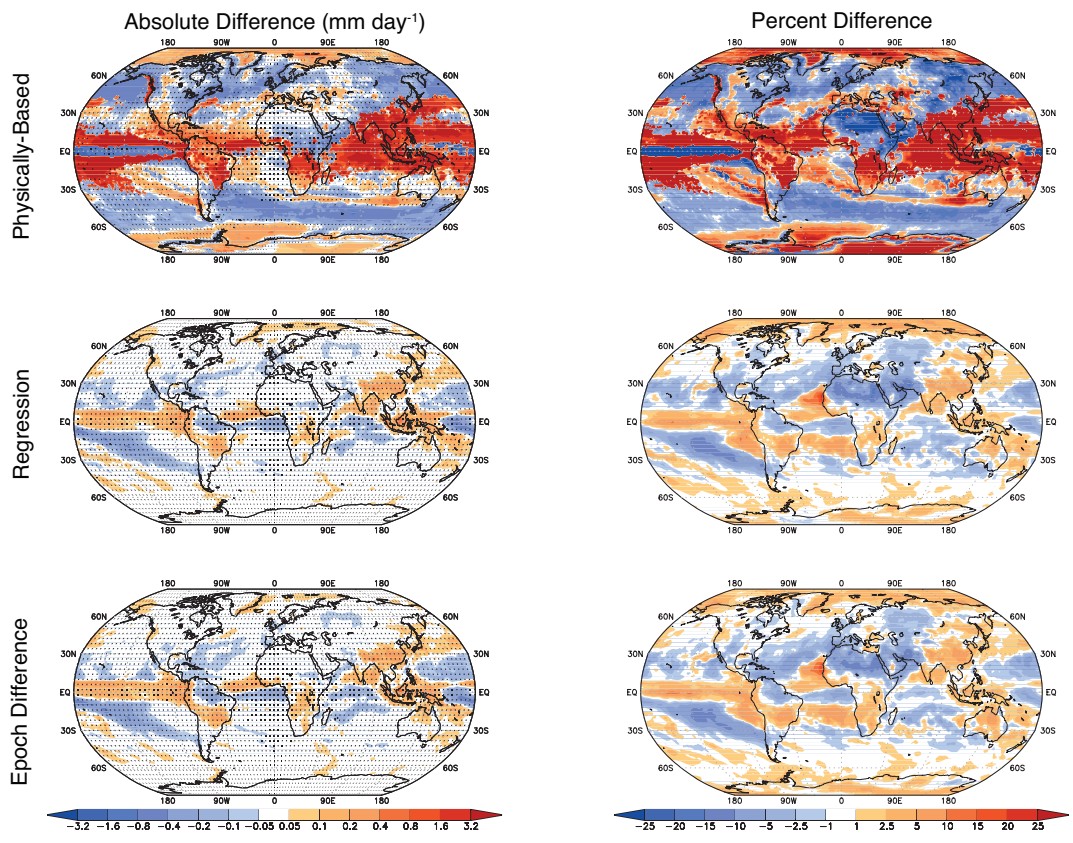

**Figure 10.** As in Figure 4 but where the reconstruction $\hat{B}$ is built on the pattern $P$ for the RCP8.5 simulation (Group 1 average over years 227–251), and global mean temperature $\Delta\bar{T}$ is averaged over years 227–251 of the RCP8.5 simulation. That is, $\hat{B} = P_{RCP8.5}(\mathbf{x})\Delta\bar{T}_{\mathrm{RCP8.5}}(227-251)$. Results shown are for the difference between the reconstruction and the actual model output $\hat{B} - B_{\mathrm{RCP8.5}}(\mathbf{x}, 227-251)$.





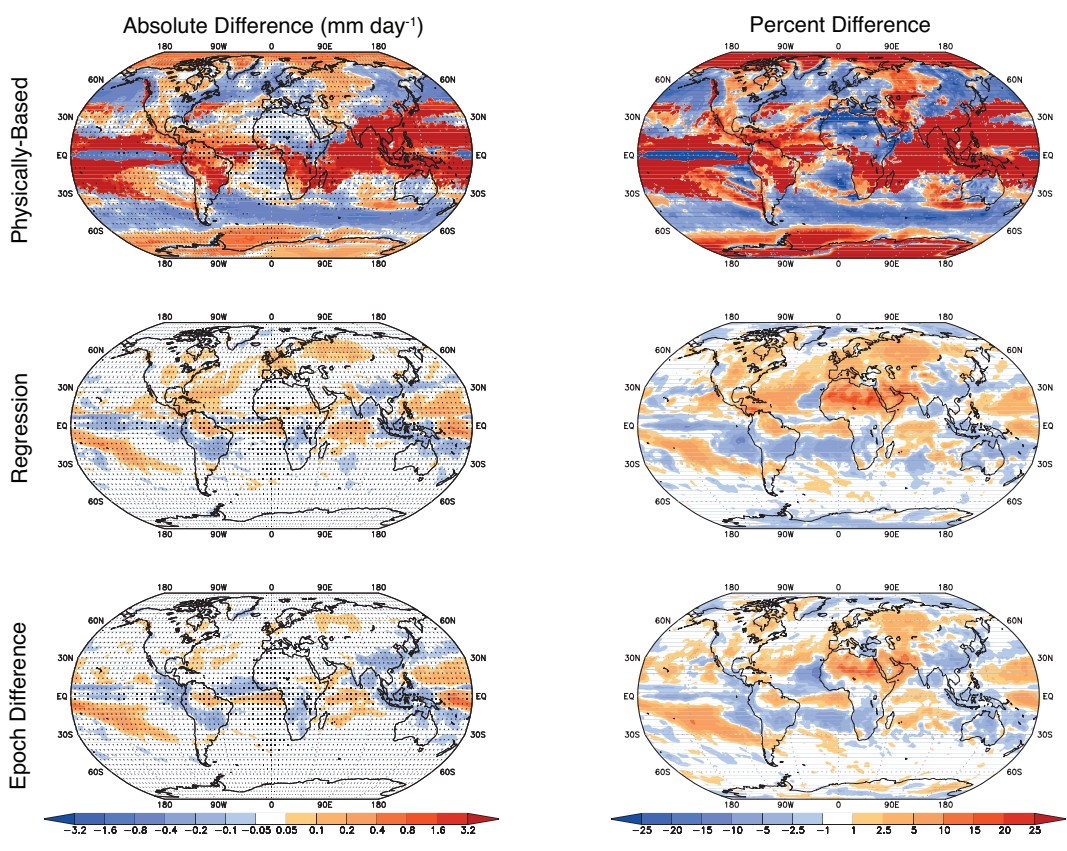

**Figure 11.** As in Figure 4 but where the reconstruction $\hat{B}$ is built on the pattern $P$ for the RCP8.5 simulation (Group 1 average over years 227–251), and global mean temperature $\Delta\bar{T}$ is averaged over years 116–140 of the RCP8.5 simulation. That is, $\hat{B} = P_{RCP8.5}(\mathbf{x})\Delta\bar{T}_{RCP8.5}(116-140)$. Results shown are for the difference between the reconstruction and the actual model output $\hat{B} - B_{RCP8.5}(\mathbf{x}, 116-140)$.





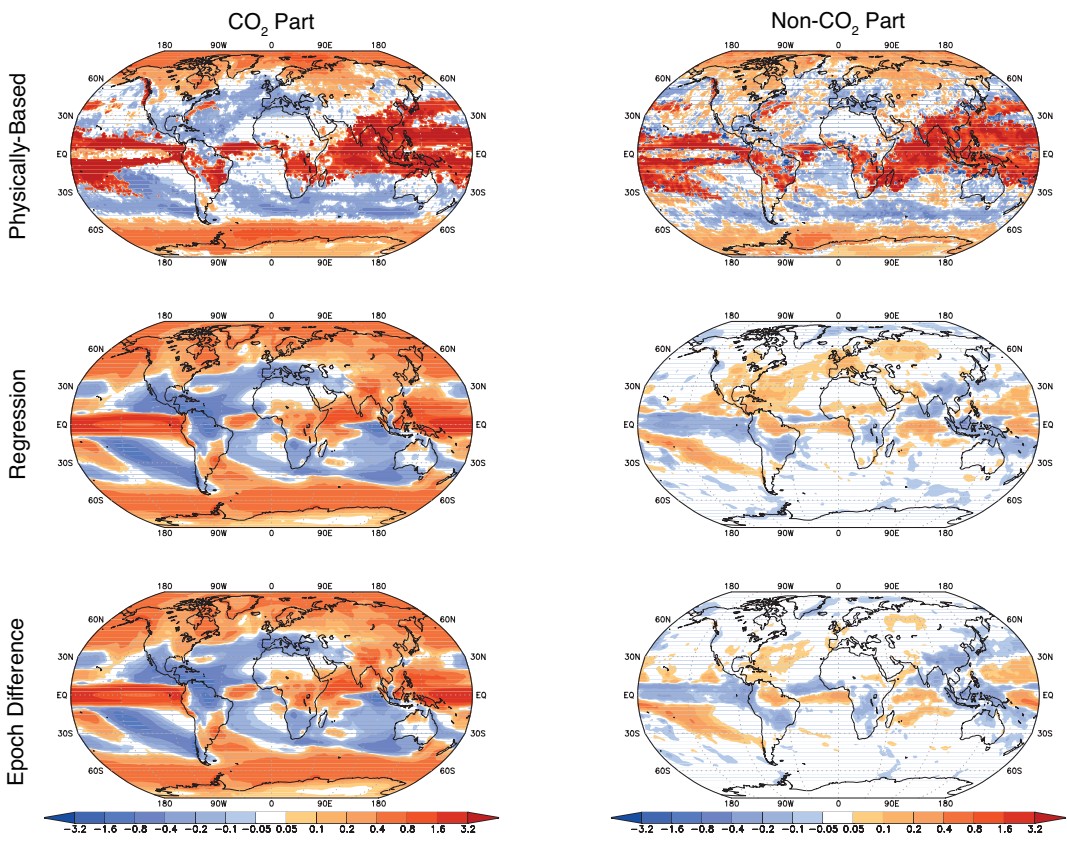

**Figure 12.** The $CO_2$ (left) and non-$CO_2$ (right) responses over the period 227–251 of the RCP8.5 simulation. $CO_2$ response is calculated as $\Delta \hat{B} = P_{1pctCO2} \bar{T}_{RCP8.5}(227 - 251)$, and non-$CO_2$ response is calculated as $\Delta \hat{B} = (P_{RCP8.5} - P_{1pctCO2}) \bar{T}_{RCP8.5}(227 - 251)$ (see Equation 1). Top row shows results for the physically-based method, middle row shows the regression method, and bottom row shows the epoch difference method.





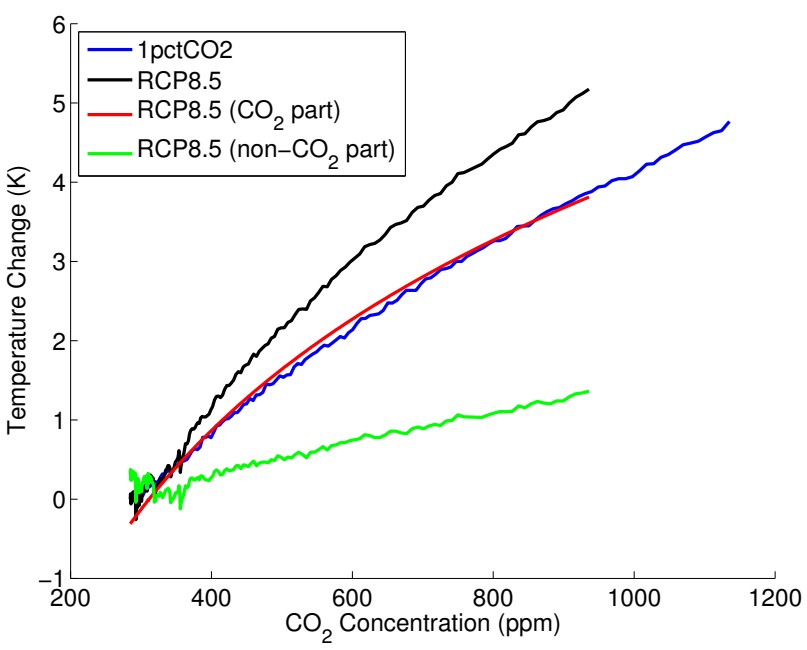

**Figure 13.** Decomposition of global mean temperature change (as a function of the $CO_2$ concentration) into its components, as described in Section 4.2.





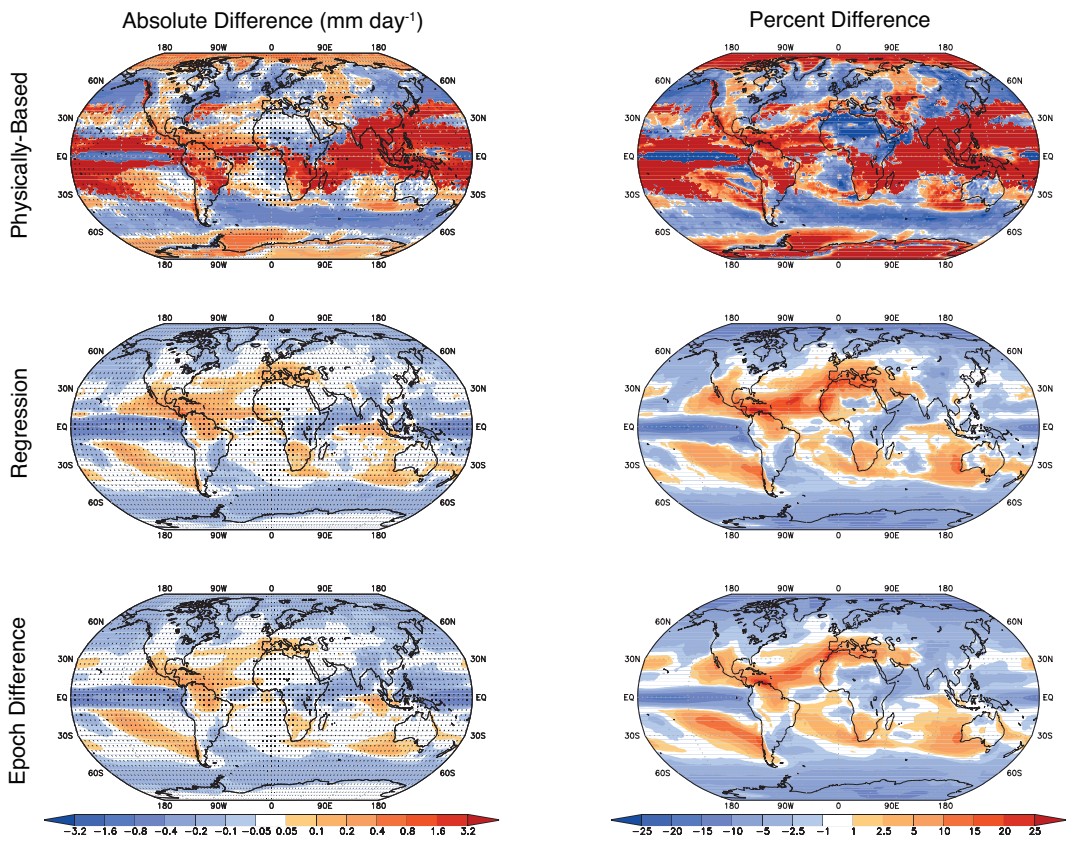

**Figure 14.** As in Figure 4 but where $\hat{B} = (P_{\mathrm{RCP8.5}} - P_{\mathrm{1pctCO2}})\bar{T}_{RCP8.5,nonCO_2}(227-251) + P_{\mathrm{1pctCO2}}\bar{T}_{RCP8.5,CO_2}(227-251)$, and results are shown for $\hat{B} - B_{\mathrm{RCP8.5}}(227-251)$. (See Equation 1 and the discussion surrounding Equation 6 for details.)





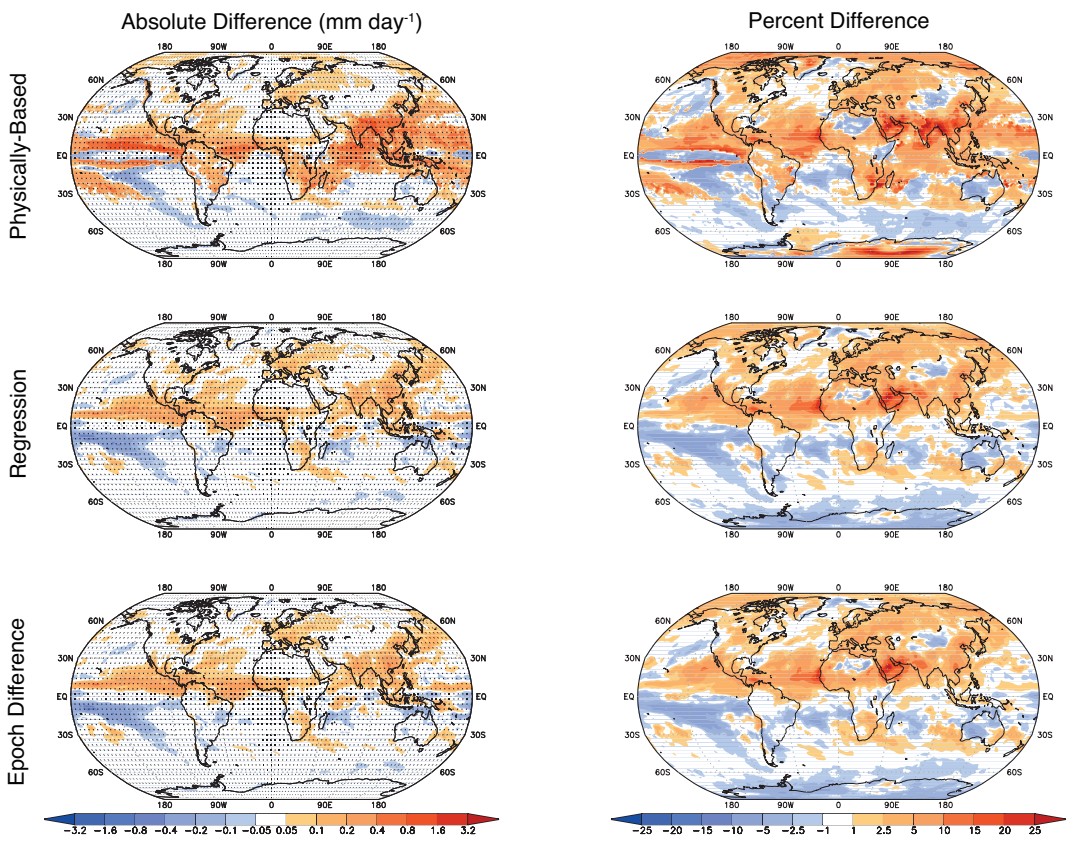

**Figure 15.** As in Figure 4 but where $\hat{B} = (P_{\text{RCP8.5}} - P_{\text{1pctCO2}})\bar{T}_{RCP8.5,nonCO_2}(116-140) + P_{\text{1pctCO2}}\bar{T}_{RCP8.5,CO_2}(116-140)$, and results are shown for $\hat{B} - B_{\text{RCP8.5}}(116-140)$. (See Equation 1 and the discussion surrounding Equation 6 for details.)





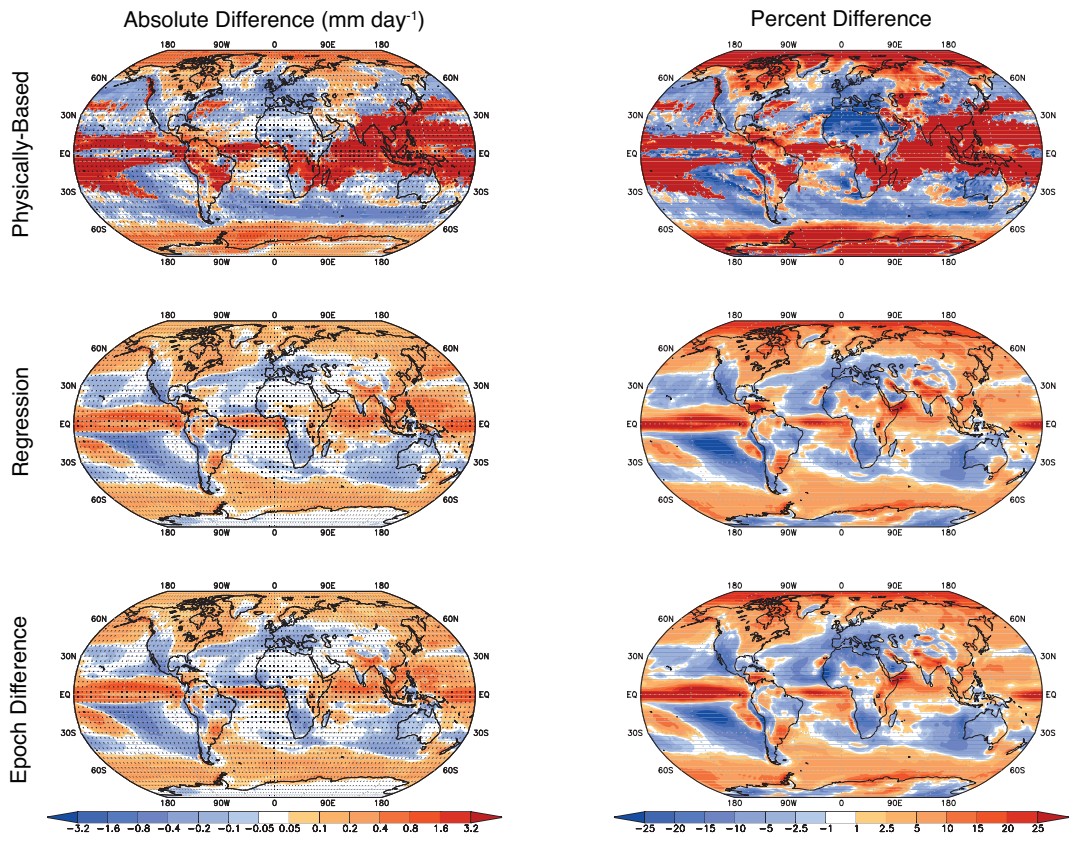

**Figure 16.** As in Figure 4 but where $\hat{B} = (P_{\mathrm{RCP8.5}} - P_{\mathrm{1pctCO2}})\bar{T}_{RCP2.6,nonCO_2}(227-251) + P_{\mathrm{1pctCO2}}\bar{T}_{RCP2.6,CO_2}(227-251)$, and results are shown for $\hat{B} - B_{\mathrm{RCP2.6}}(227-251)$. (See Equation 1 and the discussion surrounding Equation 6 for details.)





## Appendix A: Comparison of Pattern Scaling Between Two Groups of Models

Figure 17 further supports the findings in Section 3.3 by showing that the patterns $P(\mathbf{x})$ are not statistically different for Groups 1 and 2 except for isolated areas. The results for the physically-based method indicate that the findings of Lau et al. (2013) are generally reproduced here, in that the pattern is largely robust across different groups of models.

5  Figures 18 and 19 show differences in the reconstructions, averaged over years 116–140. More specifically, Figure 18 shows differences $P_{\text{Group 2}}\Delta\bar{T}_{\text{Group 1}} - \Delta B_{\text{Group 1}}$ and Figure 19 shows differences $P_{\text{Group 1}}\Delta\bar{T}_{\text{Group 2}} - \Delta B_{\text{Group 2}}$.

The results in Figures 18 and 19 for the physically-based method are both qualitatively and quantitatively similar to those in Figure 4, and global RMS values are similar. Conversely, results for the regression and epoch difference methods, while similar to each other, have qualitatively more error than the results in Figure 4. Global RMS values are 2–3 times higher in Figure 18 than in Figure 4, but errors in

10  Figure 19 are comparable to Figure 4. This might be expected, as on average, $\Delta\bar{T}_{\text{Group 1}} \approx \Delta\bar{T}_{\text{Group 2}}$, so differences in Figure 19 would be small, whereas differences in Figure 18 are driven by differences in the patterns $P_{\text{Group 1}}$ and $P_{\text{Group 2}}$ (Figure 17). As discussed in Section 3.3, the physically-based method shows some statistically significant regions of error in both Figures 18 and 19, whereas practically no region is statistically significant for the regression and epoch difference methods.





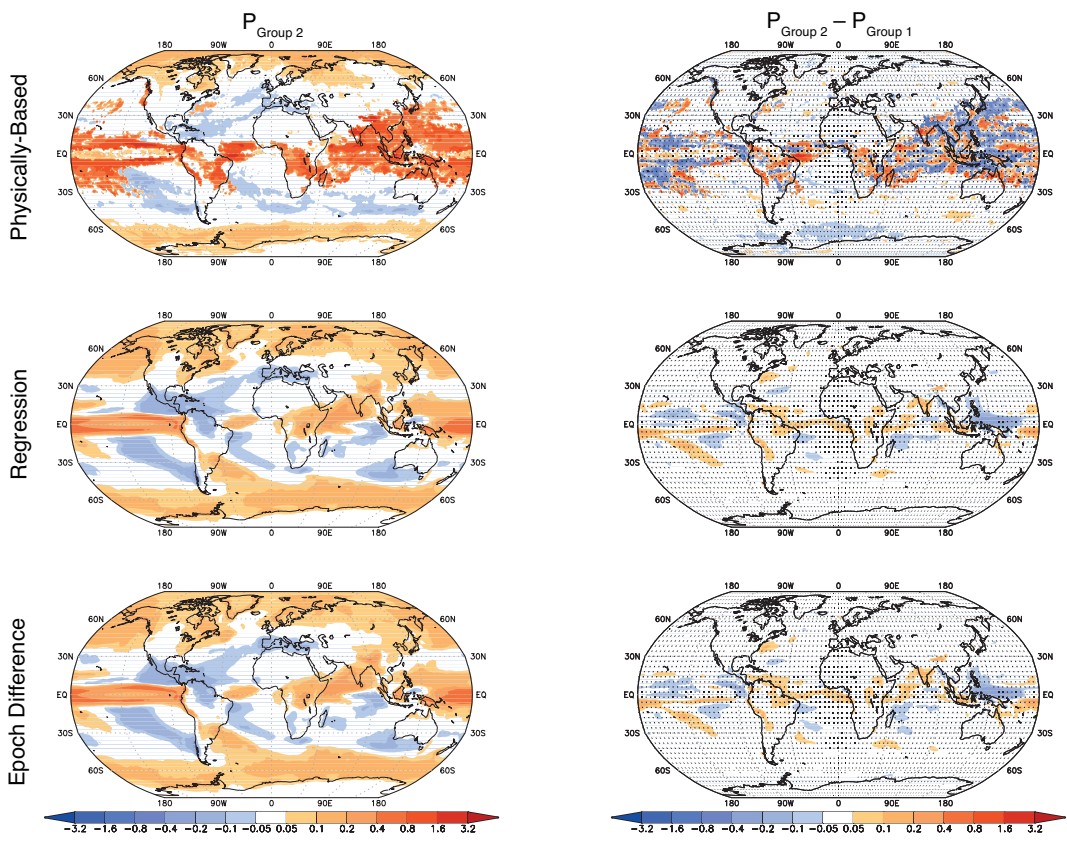

**Figure 17.** Differences in time-invariant patterns $P(\mathbf{x})$ among the two groups of models (Table 1), calculated for the 1pctCO2 simulation. Left column shows the multi-model average for Group 2, and right column shows the differences in multi-model averages among the two groups. All values shown have units mm day$^{-1}$ K$^{-1}$. Stippling indicates a lack of statistical significance in the pattern of differences (Section 2.2).





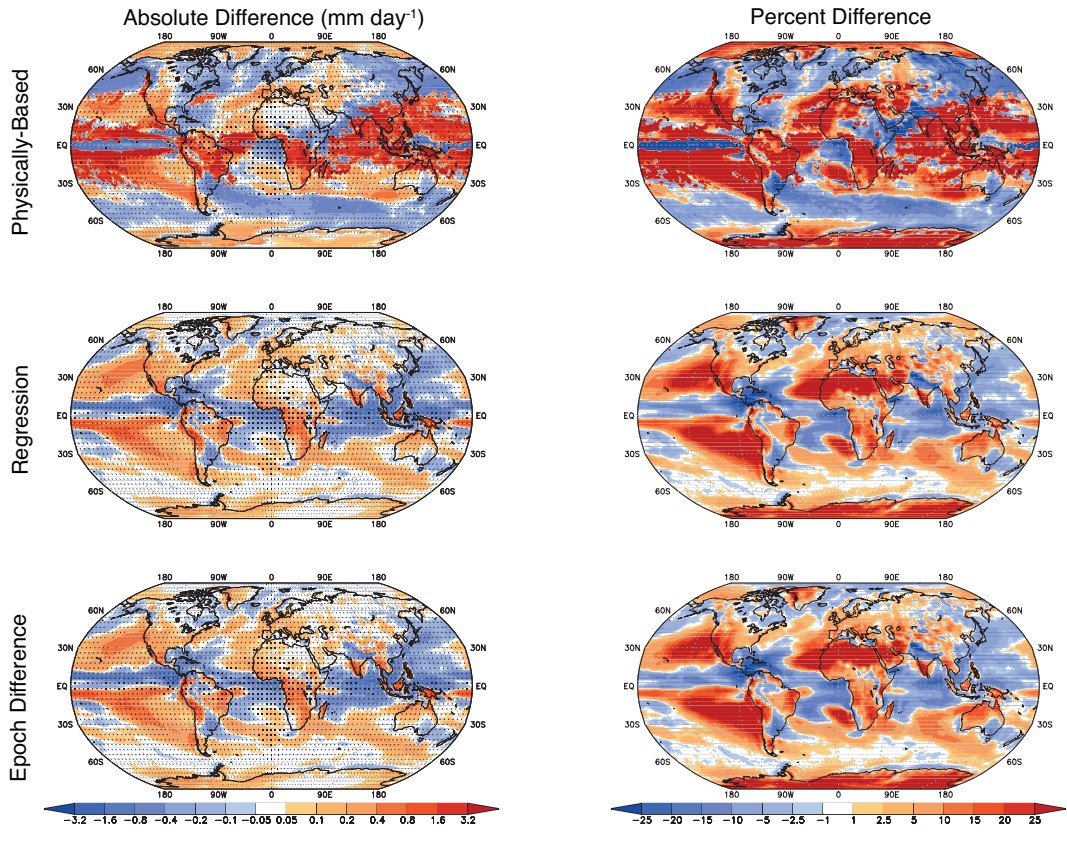

**Figure 18.** As in Figure 4 but where the reconstruction $\hat{B}$ is built on the pattern $P$ for Group 2 (average of years 116–140 of the 1pctCO2 simulation), and global mean temperature $\Delta\bar{T}$ is averaged over years 116–140 of Group 1. That is, $\hat{B} = P_{\text{Group2}}(\mathbf{x})\Delta\bar{T}_{\text{Group1}}(116-140)$. Results shown are for the difference between the reconstruction and the actual model output $\hat{B} - B_{\text{Group1}}(\mathbf{x}, 116-140)$.





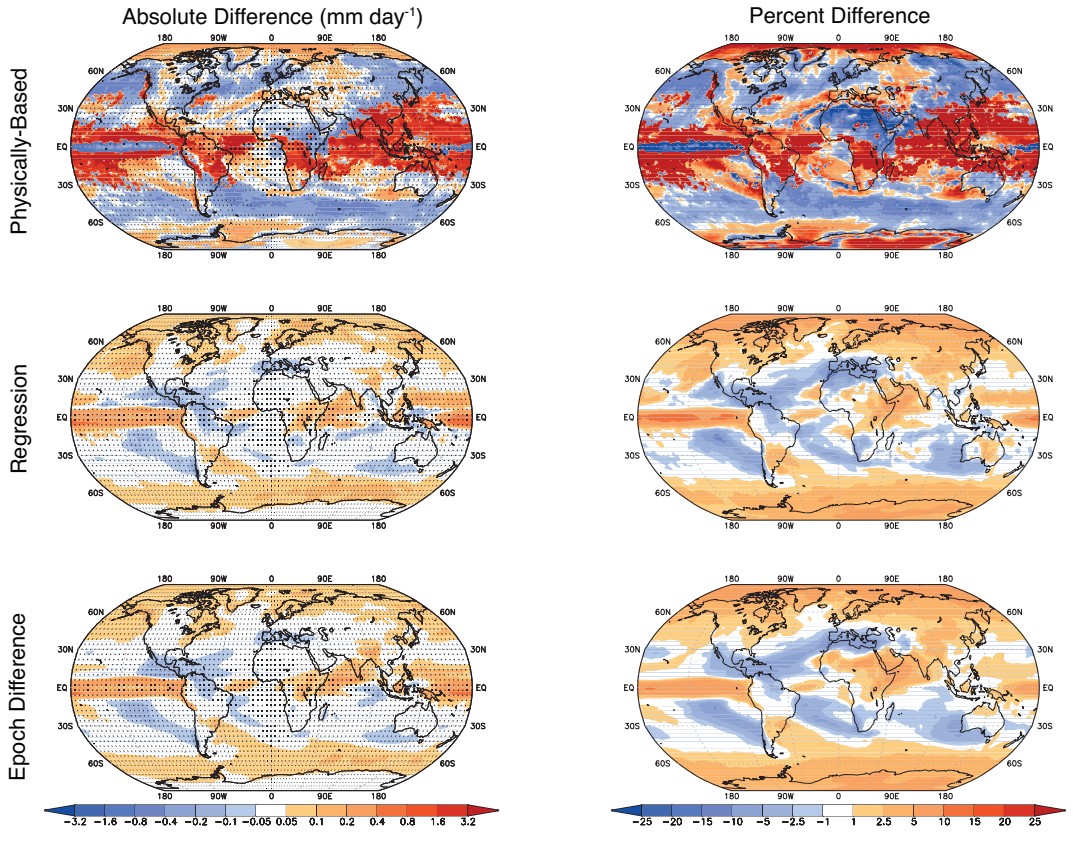

**Figure 19.** As in Figure 4 but where the reconstruction $\hat{B}$ is built on the pattern $P$ for Group 1 (average of years 116–140 of the 1pctCO2 simulation), and global mean temperature $\Delta\bar{T}$ is averaged over years 116–140 of Group 2. That is, $\hat{B} = P_{\mathrm{Group1}}(\mathbf{x})\Delta\bar{T}_{\mathrm{Group2}}(116-140)$. Results shown are for the difference between the reconstruction and the actual model output $\hat{B} - B_{\mathrm{Group2}}(\mathbf{x}, 116-140)$.