# Peer review of "Exploring precipitation pattern scaling methodologies and robustness among CMIP5 models"

_Geoscientific Model Development, 2016_

## Referee Comment (RC1)

Review comments on the manuscript

**Exploring precipitation pattern scaling methodologies and robustness among CMIP5 models**

by B. Kravitz, C. Lynch, C. Hartin and B. Bond-Lamberty

submitted to Geoscientific Model Development.

Recommendation: Accept with substantial revisions.

In my opinion, the paper constitutes an interesting contribution that should be published in Geoscientific Model Development after adequate revisions. An evaluation of the performance of different pattern scaling methods for climate variables other than temperature (here: precipitation) is of significant practical importance. The evident main conclusion of the work is that two of the methods work reasonably but the third one does not. This should be clarified in the text.

**1   General comments**

1. Regarding the two traditionally-used methods, here termed the regression and epoch methods, I mainly agree with the conclusions presented by the authors. These methods appear to be fit for scaling precipitation. However, the verbal assessments given for the "physically-based" method in the manuscript do not seem to be supported by the quantitative results presented in the figures and Table 3. Examples of statements that I find unjustified: "the physically-based method shows a greater degree of robustness (less relative root-mean-square variation than the other two methods) and could be a particularly advantageous method if outstanding biases could be reduced" (abstract, l. 7–9); "This indicates the potential for robustness of the physically-based method" (p. 5, l. 2); "The overall performance of the physically-based method is still worse in all cases, but these results suggest that if the overall bias in the physically-based method could be reduced or corrected, it holds great promise in being a useful pattern scaling method..." (p. 8, l. 26–28); "The physically-based method has substantially worse performance than the other two methods but shows some features of robustness that could be advantageous if overall biases in the method could be reduced" (p. 12, l. 16–18).

   In all the examples studied, the performance of the "physically-based" method appears to be inferior to the other two methods (Table 3). In some cases (e.g., that depicted in Fig. 6), the gap between the performances is apparently somewhat smaller. Note, however, that in these experiments the magnitude of the projected change $B$ is small, which makes the scaling error $\hat{B} - B$ small as well. Accordingly, in these cases the small RMS error produced by the "physically-based" method is likely to be a trivial consequence of the smallness of $B$. (See further discussion in "specific comments".)

   Furthermore, on l. 10–11 of p. 3 it is stated that "There are many possibilities for physically-based approaches". Therefore, I suggest that the authors should use some other, more specified name for the version of the method examined in this paper. Note also that 'physically-based' inherently sounds very positive and thus a more neutral term should be preferred; particularly, taking into account the low performance of that method.

2. The number of figures in the paper, 19, is excessive. In particular, there are plenty of figures (14) that visually very similar, consisting of a set of six global map panels. A high level of concentration is required for a reader in order to study this large manifold of illustrations. I find that it is mainly Figs. 4, 9, 10, 12, 14 and 16 that include key information. Conversely, Figs. 5–7, 11, 15 and 17–19 are not that essential and mainly relevant for readers of special interest. For the majority of readers, it would facilitate reading the article if these figures (or a significant portion of them) would be shifted into an electronic supplemental file that is available in conjunction with the article.

3. Is the precipitation variable discussed in the paper an annual mean? That should be specified in the abstract, introduction, conclusions and, perhaps, in some of the figure captions as well.

4. Compared to the other two methods, the performance of the"physically-based" method is very poor. The poorness is so striking that I recommend that the authors should check the correctness of their algorithms once again.

**2 Specific comments**

1. Interpolation, extrapolation. For readers less familiar with the idea, please specify that you are dealing with interpolation (extrapolation) **in time** (p. 1, 3, 7, 8 and 12).

2. Earth System Models (ESMs) vs. Atmosphere-Ocean General Circulation Models (AOGCMs). According to the definition applied in Chapter 9 of IPCC (2013), ESMs are those climate models that include an interactive carbon cycle. All models listed in Table 1 of your paper do not fulfil this criterion but belong to the category of AOGCMs. I recommend that you would use the same terminology as IPCC (2013). — This does not have any impact on the quantitative findings as you have used concentration-driven model runs alone (p. 5, l. 7).

3. P. 4, l. 8–17: The idea of the "physically-based" method should be explained in more detail. The present formulation is not adequate to make the idea understandable without consulting the reference.

4. There is an error in Eq. (5): in the denominator, replace $s_1^2$ by $s_1^4$ and $s_2^2$ by $s_2^4$. Check whether this is an typing error only or whether you have used the wrong $df$ in the calculations.

5. P. 7, l. 18: the poorer performance of these two methods may be due to the large contribution of noise in the pattern of $P$ that is determined from the early years of the simulation when the true climate change signal is weak.

6. P. 7, l. 20–23: The error for the "physically-based" method is not similar but nearly double that produced by the other methods (Table 3). More importantly: the smallness of the error for the"physically-based" method may have been caused by the fact that $P = 0$ over the majority of the domain. Then, in these areas $\hat{B} = 0$ as well and, since $B$ is small, the difference $B - \hat{B} = 0$ is likewise small. Thus, the smallness of the RMS error is not any indication of the good performance of the "physically-based" method. See also general comment 1 and the text that you have written on p. 8, l. 19–21 and p. 11, l. 12.

7. P. 7, l. 27–28: "error is reduced by a factor of two for the physically-based method": is this a trivial consequence of the smallness of $B$? "and increases by a factor of two for the regression and epoch difference methods": this may be an indication of true non-linearity. Also, p. 8, l. 22–26 need revision.

8. Section 3.3 and Fig. 8: When you present the results for a certain number of models, have you used in each experiment **the same** sub-ensemble models in calculating $P$ and $B$? Or are the models chosen randomly for that comparison? Please clarify.

9. P. 9, l. 17–19: I did not understand the idea. How the dominance of the $CO_2$ response helps to apply the non-$CO_2$ pattern for the other scenarios? Please clarify. Note also that the non-$CO_2$ response includes both a warming (other GHGs) and cooling component (aerosol forcing) that may have different ratios in the various RCP forcing scenarios.

10. P. 10, l. 28–32: Note that warming does not follow radiative forcing immediately but, due to the thermal inertia of oceans etc., with a lag. Has this been taken into account? If not, a caveat should be included in the text.

11. P. 34, l. 9–11: In my opinion, there is a contradiction between the text and the Figure captions 18–19. In the captions, it is stated that $P$ is extracted from one group of models and $\Delta T$ and $B$ from the another group. Thus, the experiments would be "antisymmetric" and accordingly, one would expect that the errors would be of a similar order of magnitude. Differences in $P$ between the groups 1 and 2 should affect Figs. 18 and 19 by about a similar magnitude. According to the text, figures and Table 3, however, this is not so. Please check and clarify.

12. The discussion presented in the Appendix might be transferred into electronic supplementary material.

**3  Minor comments**

1. P. 1, l. 12–13: for other models -> to be utilized in other models ?

2. P. 2, l. 19: a long history of research -> a fairly long history of research (the method has been used for a few decades, not millenia).

3. P. 2, l. 26–27: "no single fit (e.g., regression coefficients) will be applicable to all grid points" (and a similar statement on p. 3, l. 27–28). This is a trivial consequence of the fact that the modelled precipitation change is not geographically uniform. If you want to say something more, please clarify.

4. P. 3, l. 6–7: "If the climate response is perfectly linear, then any pattern scaling method will work equally well and will be highly accurate." I would prefer a more conditional formulation, e.g.: "If the climate response were perfectly linear, then any pattern scaling method would work equally well and would be highly accurate."

5. P. 3., l. 9: Conversely -> In principle; the findings of the present work do not favour the "physically-based" method.

6. P. 3, l. 30: "this approach automatically accounts for correlations between local temperature and local precipitation changes". How?

7. P. 4, l. 30: This may be caused (i) by the rather small area of the polar regions and (ii) by the fact that both $B$ and $\hat{B}$ are relatively small there.

8. P. 7, 7–8: "If the scaling pattern $P(x)$ truly is time-invariant, then the results presented in this section will be identical to those previously discussed." -> "If the scaling pattern $P(x)$ truly were time-invariant, then the results presented in this section would be identical to those previously discussed." (They are not identical.)

9. P. 7, l. 16: none -> virtually none ?

10. P. 8, l. 9: I did not understand "The values in Table 3 indicate that Group 1 (13 models) is not an outlier." Please clarify.

11. P. 8, l. 27: I do not agree with "holds great promise".

12. Title of section 4 might be modified: you discuss the total forcing and its partition into the $CO_2$ and non-$CO_2$ components.

13. P. 9, l. 17: "There is no a priori reason to expect this will work". Do you mean "There is no a priori reason to expect **that** this will work"?

14. P. 9, l. 30: Giving the actual years (e.g., 2076–2100 for model years 227–251?) would be informative (in figure captions as well).

15. P. 10, l. 11–12: One possible explanation is aerosol forcing.

16. P. 10, 27–28: "If the approach fails, it is because this pattern does not represent actual non-$CO_2$ forcing." Noise due to unforced internal variability in the climate system may also have an influence.

17. P. 10, l. 30: $\log_2([CO_2])$. In what units $[CO_2]$ is expressed? This determines the values of the coefficients.

18. P. 11, l. 1–2: "The temperature contribution of the non-CO2 part increases with the CO2 concentration" was not entirely clear for me.

19. P. 11, l. 22–23: "The epoch difference and regression methods show too much CO2 response and not enough non-CO2 response as indicated by the patterns displayed in Figure 15." Would you please explain in more detail how one can see this?

20. P. 11, l. 27: "values depicted in Figure 16 are almost entirely due to CO2 forcing". According to Table 2, the ratio of non-$CO_2$ to $CO_2$ forcing does not differ substantially between these two RCP scenarios.

21. P. 11, l. 28–29: "this indicates that the non-CO2 forcing in RCP2.6 is insufficiently large to overcome issues with low signal-to-noise ratios in reconstructing patterns of precipitation change using this sort of decomposition." This was difficult to understand. Please explain in more detail.

22. P. 11, l. 30: "polar amplification **of the precipitation response**". In general, polar amplification refers to the temperature response.

23. P. 12, l. 18: the methods work relatively well, but i regard "excellent" as a too emphatic word.

24. P. 12, l. 25–26: "it is the best equipped to deal with these sources of nonlinearity." Perhaps in theory, but the present findings do not support this statement.

25. P. 12, l. 32 – p. 13, l. 1: "However, given the difficulties many Earth System Models have with proper representations of convective processes and the resulting precipitation biases those difficulties cause.." -> "However, given the difficulties **that** many Earth System Models (-> **climate models (?)**) have with proper representations of convective processes and the resulting precipitation biases **that** those difficulties cause.." (would be much more easy to understand for a non-native reader).

26. Caption of Fig. 3: should there be $\hat{B}$ rather than $\hat{T}$?

27. Caption of Fig. 5: "Differences in the precipitation scaling pattern..." Actually, the left column panels do not depict differences but the absolute distributions of $P_{1-50}$. Caption text needs revision. The same error occurs in the captions of Figs. 9 and 17.

28. Fig. 8: Should the title of the top panel be "physically-based" rather than "reconstruction"?

29. Fig. 11: this period, years 1965–1989 (if I have calculated correctly), actually does not yet belong to the RCP but to the historical period of the CMIP5 runs.

**References**

IPCC, 2013: *Climate Change 2013: The physical science basis. Contribution of Working Group I to the Fifth Assessment Report of the Intergovernmental Panel on Climate Change.* Cambridge University Press, Cambridge, U.K., 1535 pp, [Stocker, T.F., D. Qin, G.-K. Plattner, M. Tignor, S.K. Allen, J. Boschung, A. Nauels, Y. Xia, V. Bex and P.M. Midgley (eds.)].

---

## Referee Comment (RC2) · Anonymous Referee #2 · 14 Jan 2017

The submitted manuscripts compares several methods for the pattern scaling of precipitation across time periods and scenarios. They compare a regression based approach, an epoch difference and a 'physically' approach. I cannot recommend this paper for publication because of two significant errors in the methodology, combined with a manuscript which is too long, without a clear structure.

Firstly, the 'physically-based' approach, which is based on the work of Lau (2013), is very likely incorrectly applied. In Figure 4, which is basically a test of whether the methods are able to reconstruct an in-sample pattern of precipitation using the same ensemble and time period as a test response pattern as was used to produce the pattern itself. In this case, the method produces errors an order of magnitude greater than the other approaches - which suggests that there is an error in application. If there is no error, this huge discrepancy requires an explanation.

However, even taking this into account, there is little logic that this approach is 'physically-based' at all. The precipitation rates are binned by different monthly rain rates, averaged over the ensemble and recombined into a single pattern. If a single pattern is being scaled - the ability to treat differently rain rates in different regimes has already been lost. The entire concept is not clearly defensible.

The separation of response patterns into CO2 and non-CO2 components could potentially be useful, but the implementation is flawed. The authors assume in Figure 14 that the non-CO2 response pattern is given by the difference between the RCP8.5 and 1pctCO2 patterns. This is not correct.

Assume there is a 'pure CO2' precipitation response which can be measured from the 1pctCO2 simulation:

 $B_{CO2} = \Delta P_{1pctCO2} / \Delta T_{1pctCO2}$

If we assume things are linear, the precipitation response in RCP8.5 is this pure CO2 response, multiplied by the pure CO2 warming, plus a non-CO2 response:

 $\Delta P_{RCP85} = \Delta T_{RCP85,CO2} B_{CO2} + \Delta T_{RCP85,nonCO2} B_{nonCO2}$

so - by solving this, we get the  $B_{nonCO2}$  pattern and could reconstruct the  $\Delta P_{RCP85}$  exactly.

However, it's still not clear that CO2/nonCO2 is the correct way to break this problem down. The nonCO2 component is a broadly mix of aerosols, and other greenhouse gases (CH4, N2O etc). These two groups can have opposite effects on global mean temperature - potentially making  $\Delta T_{RCP85,nonCO2}$  near zero and making the above equation ill-posed.

Furthermore, CH4 and aerosols have very different precipitation response fingerprints. RCP8.5 and RCP2.6 have very similar aerosol forcings, but very different CH4 trajectories, so the nonCO2 pattern appropriate for RCP8.5 would be very different than that for RCP2.6.

A far more logical decomposition would be between GHG and nonGHG forcing. The authors could solve this by treating the 1pctCO2 response as the GHG response pattern, and then in RCP8.5 calculating the effective CO2 concentration using the emission factors for each of the non CO2 gases, and then computing the  $\Delta T_{RCP85,GHG}$  as before using effCO2 rather than CO2 itself.

The general formulation of the rest of the paper, and the treatment of the other two pattern scaling approaches, is broadly correct - but the presentation is often frustratingly vague. It is often not made clear what is in sample, and what is being tested. In Figure 8, are the same models being used to make the patterns and the test the errors? In Figure 11, is it 1pctCO2 or RCP85 being reconstructed?

The authors should correct the major errors above and restructure the paper to ensure concise and clear communication before resubmission.

Lau, W. K.-M., Wu, H.-T., and Kim, K.-M.: A canonical response of precipitation characteristics to global warming from CMIP5 models, Geophys. Res. Lett., 40, 3163–3169, doi:10.1002/grl.50420, 2013.

СЗ

---

## Author Comment (AC1)

Exploring precipitation pattern scaling methodologies and robustness among CMIP5 models
Kravitz et al., Geoscientific Model Development
Response to reviewers

Reviewer comments in plain text. **Responses in bold.**
* * *
**General response**

**We thank the reviewers for their comments on our paper. Both reviewers were critical of the "physically-based" method, and we have carefully considered their points. We agree that it is important to evaluate this method more carefully, including additional checks of the accuracy of our implementation. Exploring this method would also require a discussion of its usefulness as a pattern scaling method and why we obtained the results that we did. Given the large increase in scope this would require, which would distract from our assessments of the performance of the other two methods, we have elected to remove mention of this method from the present manuscript. We will do a better job with it in a future study.**
* * *
Reviewer #1

In my opinion, the paper constitutes an interesting contribution that should be published in Geoscientific Model Development after adequate revisions. An evaluation of the performance of different pattern scaling methods for climate variables other than temperature (here: precipitation) is of significant practical importance. The evident main conclusion of the work is that two of the methods work reasonably but the third one does not. This should be clarified in the text.

**We thank the reviewer for his/her comments. As stated above, we have removed the third method, so our conclusions will change slightly. We have updated the text to accommodate this.**

General comments

Regarding the two traditionally-used methods, here termed the regression and epoch methods, I mainly agree with the conclusions presented by the authors. These methods appear to be fit for scaling precipitation. However, the verbal assessments given for the "physically-based" method in the manuscript do not seem to be supported by the quantitative results presented in the figures and Table 3. Examples of statements that I find unjustified: "the physically-based method shows a greater degree of robustness (less relative root-mean-square variation than the other two methods) and could be a particularly advantageous method if outstanding biases

could be reduced" (abstract, l. 7–9); "This indicates the potential for robustness of the physically-based method" (p. 5, l. 2); "The overall performance of the physically-based method is still worse in all cases, but these results suggest that if the overall bias in the physically-based method could be reduced or corrected, it holds great promise in being a useful pattern scaling method..." (p. 8, l. 26–28); "The physically-based method has substantially worse performance than the other two methods but shows some features of robustness that could be advantageous if overall biases in the method could be reduced" (p. 12, l. 16–18).

In all the examples studied, the performance of the "physically-based" method appears to be inferior to the other two methods (Table 3). In some cases (e.g., that depicted in Fig. 6), the gap between the performances is apparently somewhat smaller. Note, however, that in these experiments the magnitude of the projected change B is small, which makes the scaling error $\hat{B}$ – B small as well. Accordingly, in these cases the small RMS error produced by the "physically-based" method is likely to be a trivial consequence of the smallness of B. (See further discussion in "specific comments".)

Furthermore, on l. 10–11 of p. 3 it is stated that "There are many possibilities for physically-based approaches". Therefore, I suggest that the authors should use some other, more specified name for the version of the method examined in this paper. Note also that 'physically-based' inherently sounds very positive and thus a more neutral term should be preferred; particularly, taking into account the low performance of that method.

**We agree with all of the points in the previous several paragraphs of the reviewer's assessment. Per the general response above, we have removed the physically-based method from this manuscript and all text associated with it.**

The number of figures in the paper, 19, is excessive. In particular, there are plenty of figures (14) that visually very similar, consisting of a set of six global map panels. A high level of concentration is required for a reader in order to study this large manifold of illustrations. I find that it is mainly Figs. 4, 9, 10, 12, 14 and 16 that include key information. Conversely, Figs. 5–7, 11, 15 and 17–19 are not that essential and mainly relevant for readers of special interest. For the majority of readers, it would facilitate reading the article if these figures (or a significant portion of them) would be shifted into an electronic supplemental file that is available in conjunction with the article.

**We agree with the reviewer that there were too many figures. After reviewing the paper, we have moved Figures 5-7 and 17-19 to supplemental material. We have also removed Figure 2.**

Is the precipitation variable discussed in the paper an annual mean? That should be specified in the abstract, introduction, conclusions and, perhaps, in some of the

figure captions as well.

**Agreed.  We have added mentions of this throughout the paper.**

Compared to the other two methods, the performance of the "physically-based" method is very poor. The poorness is so striking that I recommend that the authors should check the correctness of their algorithms once again.

**Per the general response above, we have removed the physically-based method from this manuscript.**

Specific comments

Interpolation, extrapolation. For readers less familiar with the idea, please specify that you are dealing with interpolation (extrapolation) in time (p. 1, 3, 7, 8 and 12).

**Thanks for pointing that out.  We have added more specificity where appropriate.**

Earth System Models (ESMs) vs. Atmosphere-Ocean General Circulation Models (AOGCMs). According to the definition applied in Chapter 9 of IPCC (2013), ESMs are those climate models that include an interactive carbon cycle. All models listed in Table 1 of your paper do not fulfill this criterion but belong to the category of AOGCMs. I recommend that you would use the same terminology as IPCC (2013). — This does not have any impact on the quantitative findings as you have used concentration-driven model runs alone (p. 5, l. 7).

**A point well taken.  We have replaced all mentions of ESM with AOGCM in the manuscript.**

P. 4, l. 8–17: The idea of the "physically-based" method should be explained in more detail. The present formulation is not adequate to make the idea understandable without consulting the reference.

**Per the general response, we have removed mentions of the physically based method.**

There is an error in Eq. (5): in the denominator, replace $s^2_1$ by $s^4_1$ and $s^2$ by $s^4_2$. Check whether this is an typing error only or whether you have used the wrong df in the calculations.

**This was just a typo in the manuscript.  Thanks for pointing that out.**

P. 7, l. 18: the poorer performance of these two methods may be due to the large contribution of noise in the pattern of P that is determined from the early years of

the simulation when the true climate change signal is weak.

**Good point. We have added a sentence to this effect.**

P. 7, l. 20–23: The error for the "physically-based" method is not similar but nearly double that produced by the other methods (Table 3). More importantly: the smallness of the error for the"physically-based" method may have been caused by the fact that P = 0 over the majority of the domain. Then, in these areas $\hat{B}$ = 0 as well and, since B is small, the difference $B - \hat{B}$ = 0 is likewise small. Thus, the smallness of the RMS error is not any indication of the good performance of the "physically-based" method. See also general comment 1 and the text that you have written on p. 8, l. 19–21 and p. 11, l. 12.

**Per the general response, we have removed mentions of the physically based method.**

P. 7, l. 27–28: "error is reduced by a factor of two for the physically-based method": is this a trivial consequence of the smallness of B? "and increases by a factor of two for the regression and epoch difference methods": this may be an indication of true non-linearity. Also, p. 8, l. 22–26 need revision.

**Per the general response, we have removed mentions of the physically based method. We have revised what's left of the lines on page 8 to improve clarity.**

Section 3.3 and Fig. 8: When you present the results for a certain number of models, have you used in each experiment the same sub-ensemble models in calculating P and B? Or are the models chosen randomly for that comparison? Please clarify.

**The models are chosen randomly. We have clarified this in the text.**

P. 9, l. 17–19: I did not understand the idea. How the dominance of the $CO_2$ response helps to apply the non-$CO_2$ pattern for the other scenarios? Please clarify. Note also that the non-$CO_2$ response includes both a warming (other GHGs) and cooling component (aerosol forcing) that may have different ratios in the various RCP forcing scenarios.

**We have removed that part of the sentence that perplexed the reviewer. As to the other point, that is well taken. We have added an additional paragraph that discusses many of these issues.**

P. 10, l. 28–32: Note that warming does not follow radiative forcing immediately but, due to the thermal inertia of oceans etc., with a lag. Has this been taken into account? If not, a caveat should be included in the text.

**We have not accounted for lags like this. We have added a caveat to the text.**

P. 34, l. 9–11: In my opinion, there is a contradiction between the text and the Figure captions 18–19. In the captions, it is stated that P is extracted from one group of models and ΔT and B from the other group. Thus, the experiments would be "antisymmetric" and accordingly, one would expect that the errors would be of a similar order of magni- tude. Differences in P between the groups 1 and 2 should affect Figs. 18 and 19 by about a similar magnitude. According to the text, figures and Table 3, however, this is not so. Please check and clarify.

**We apologize for the confusion. We had a typo in the text, so the descriptions of the two figures appeared to be antisymmetric, but they weren't. We have fixed this so the paper better says what we actually did.**

The discussion presented in the Appendix might be transferred into electronic supplementary material.

**Agreed. We have moved this text and the associated figures to supplemental material.**

Minor comments

P. 1, l. 12–13: for other models -> to be utilized in other models ?

**Agreed and changed in the manuscript.**

P. 2, l. 19: a long history of research -> a fairly long history of research (the method has been used for a few decades, not millenia).

**Agreed and changed in the manuscript.**

P. 2, l. 26–27: "no single fit (e.g., regression coefficients) will be applicable to all grid points" (and a similar statement on p. 3, l. 27–28). This is a trivial consequence of the fact that the modelled precipitation change is not geographically uniform. If you want to say something more, please clarify.

**We agree with the reviewer's statement. We are simply reviewing what previous studies have shown.**

P. 3, l. 6–7: "If the climate response is perfectly linear, then any pattern scaling method will work equally well and will be highly accurate." I would prefer a more conditional formulation, e.g.: "If the climate response were perfectly linear, then any pattern scaling method would work equally well and would be highly accurate."

**Agreed and changed in the manuscript.**

P. 3., l. 9: Conversely -> In principle; the findings of the present work do not favour the "physically-based" method.

**We have removed mentions of the physically-based method in this manuscript.**

P. 3, l. 30: "this approach automatically accounts for correlations between local temperature and local precipitation changes". How?

**This sentence ended up being more confusing than illuminating, so we have removed it.**

P. 4, l. 30: This may be caused (i) by the rather small area of the polar regions and (ii) by the fact that both B and $\hat{B}$ are relatively small there.

**Per the general response above, we have removed this paragraph.**

P. 7, 7–8: "If the scaling pattern P (x) truly is time-invariant, then the results presented in this section will be identical to those previously discussed." -> "If the scaling pattern P (x) truly were time-invariant, then the results presented in this section would be identical to those previously discussed." (They are not identical.)

**Changed.  Thanks for the phrasing.**

P. 7, l. 16: none -> virtually none ?

**Agreed and changed in the manuscript.**

P. 8, l. 9: I did not understand "The values in Table 3 indicate that Group 1 (13 models) is not an outlier." Please clarify.

**We have clarified this sentence.**

P. 8, l. 27: I do not agree with "holds great promise".

**Agreed.  Keeping with the general response above, we have removed this paragraph.**

Title of section 4 might be modified: you discuss the total forcing and its partition into the $CO_2$ and non-$CO_2$ components.

**Changed to "Pattern Scaling for Additional Forcings".**

P. 9, l. 17: "There is no a priori reason to expect this will work". Do you mean "There is   no a priori reason to expect that this will work"?

**Yes, changed.**

P. 9, l. 30: Giving the actual years (e.g., 2076–2100 for model years 227–251?) would be   informative (in figure captions as well).

**Agreed.  Changed in the text and the relevant figure captions.**

P. 10, l. 11–12: One possible explanation is aerosol forcing.

**Agreed.  We have added a mention of aerosol forcing.**

P. 10, 27–28: "If the approach fails, it is because this pattern does not represent actual non-$CO_2$ forcing." Noise due to unforced internal variability in the climate system may also have an influence.

**Good point.  We have added this.**

P. 10, l. 30: $\log_2([CO_2])$. In what units $[CO_2]$ is expressed? This determines the values of the coefficients.

**Added units of ppmv.**

P. 11, l. 1–2: "The temperature contribution of the non-CO2 part increases with the CO2 concentration" was not entirely clear for me.

**We have revised this sentence to be less confusing.**

P. 11, l. 22–23: "The epoch difference and regression methods show too much CO2 response and not enough non-CO2 response as indicated by the patterns displayed in Figure 15." Would you please explain in more detail how one can see this?

**We have substantially revised this section.  The comment now references removed text.**

P. 11, l. 27: "values depicted in Figure 16 are almost entirely due to CO2 forcing". According to Table 2, the ratio of non-$CO_2$ to $CO_2$ forcing does not differ substantially between these two RCP scenarios.

**We have substantially revised this section.  The comment now references removed text.**

P. 11, l. 28–29: "this indicates that the non-CO2 forcing in RCP2.6 is insufficiently large to overcome issues with low signal-to-noise ratios in reconstructing patterns of precipitation change using this sort of decomposition." This was difficult to

understand. Please explain in more detail.

**We have substantially revised this section. The comment now references removed text.**

P. 11, l. 30: "polar amplification of the precipitation response". In general, polar amplification refers to the temperature response.

**Added. Thanks!**

P. 12, l. 18: the methods work relatively well, but i regard "excellent" as a too emphatic word.

**Agreed. We have rephrased this sentence.**

P.12,l.25–26:"it is the best equipped to deal with these sources of nonlinearity." Perhaps in theory, but the present findings do not support this statement.

**Agreed. We have removed the physically based method, so this sentence has been removed as well.**

P. 12, l. 32 – p. 13, l. 1: "However, given the difficulties many Earth System Models have with proper representations of convective processes and the resulting precipitation biases those difficulties cause.." -> "However, given the difficulties that many Earth System Models (-> climate models (?)) have with proper representations of convective processes and the resulting precipitation biases that those difficulties cause.." (would be much more easy to understand for a non-native reader).

**Changed. Thanks for the suggested phrasing!**

Caption of Fig. 3: should there be $\hat{B}$ rather than $\hat{T}$?

**Yes, fixed.**

Caption of Fig. 5: "Differences in the precipitation scaling pattern..." Actually, the left column panels do not depict differences but the absolute distributions of $P_{1-50}$. Caption text needs revision. The same error occurs in the captions of Figs. 9 and 17.

**Agreed. Thanks for catching that.**

Fig. 8: Should the title of the top panel be "physically-based" rather than "reconstruction"?

**Yes, thanks for catching that. Although we have removed this panel anyway, as we no longer include the physically-based reconstruction.**

Fig. 11: this period, years 1965–1989 (if I have calculated correctly), actually does not yet belong to the RCP but to the historical period of the CMIP5 runs.

**This is correct. We have clarified what we meant.**

---

## Author Comment (AC2)

Exploring precipitation pattern scaling methodologies and robustness among CMIP5 models
Kravitz et al., Geoscientific Model Development
Response to reviewers

Reviewer comments in plain text.  **Responses in bold.**
* * *
**General response**

**We thank the reviewers for their comments on our paper.  Both reviewers were critical of the "physically-based" method, and we have carefully considered their points.  We agree that it is important to evaluate this method more carefully, including additional checks of the accuracy of our implementation.  Exploring this method would also require a discussion of its usefulness as a pattern scaling method and why we obtained the results that we did.  Given the large increase in scope this would require, which would distract from our assessments of the performance of the other two methods, we have elected to remove mention of this method from the present manuscript.  We will do a better job with it in a future study.**
* * *
Reviewer #2

The submitted manuscript compares several methods for the pattern scaling of precipitation across time periods and scenarios. They compare a regression based approach, an epoch difference and a 'physically' approach. I cannot recommend this paper for publication because of two significant errors in the methodology, combined with a manuscript which is too long, without a clear structure.

**We have substantially shortened the paper and provided outlining and clearer desciptions as to our main findings.**

Firstly, the 'physically-based' approach, which is based on the work of Lau (2013), is very likely incorrectly applied. In Figure 4, which is basically a test of whether the methods are able to reconstruct an in-sample pattern of precipitation using the same ensemble and time period as a test response pattern as was used to produce the pattern itself. In this case, the method produces errors an order of magnitude greater than the other approaches - which suggests that there is an error in application. If there is no error, this huge discrepancy requires an explanation.

However, even taking this into account, there is little logic that this approach is 'physically-based' at all. The precipitation rates are binned by different monthly rain rates, averaged over the ensemble and recombined into a single pattern. If a single pattern is being scaled - the ability to treat differently rain rates in different regimes

has already been lost. The entire concept is not clearly defensible.

**After careful consideration (see general response above), we have removed the physically-based method from this manuscript.**

The separation of response patterns into CO2 and non-CO2 components could potentially be useful, but the implementation is flawed. The authors assume in Figure 14 that the non-CO2 response pattern is given by the difference between the RCP8.5 and 1pctCO2 patterns. This is not correct.

Assume there is a 'pure CO2' precipitation response which can be measured from the 1pctCO2 simulation:

$B_{CO2} = \Delta P_{1pctCO2}/\Delta T_{1pctCO2}$  If we assume things are linear, the precipitation response in RCP8.5 is this pure CO2 response, multiplied by the pure CO2 warming, plus a non-CO2 response:

$\Delta P_{RCP}85 = \Delta T_{RCP}85,_{CO2}B_{CO2} + \Delta T_{RCP}85,_{nonCO2}B_{nonCO2}$  so - by solving this, we get the $B_{nonCO2}$ pattern and could reconstruct the $\Delta P_{RCP}85$ exactly.

**We thank the reviewer for this comment. We agree that we were not as careful as we should have been in the previous iteration of this manuscript. We have added Supplemental Section 1, which goes through this derivation and arrives at a more accurate formulation for the non-CO2 pattern.**

However, it's still not clear that CO2/nonCO2 is the correct way to break this problem down. The nonCO2 component is a broadly mix of aerosols, and other greenhouse gases (CH4, N2O etc). These two groups can have opposite effects on global mean temperature - potentially making $\Delta T_{RCP}85,_{nonCO2}$ near zero and making the above equation ill-posed.

Furthermore, CH4 and aerosols have very different precipitation response fingerprints. RCP8.5 and RCP2.6 have very similar aerosol forcings, but very different CH4 trajectories, so the nonCO2 pattern appropriate for RCP8.5 would be very different than that for RCP2.6.

A far more logical decomposition would be between GHG and nonGHG forcing. The authors could solve this by treating the 1pctCO2 response as the GHG response pattern, and then in RCP8.5 calculating the effective CO2 concentration using the emission factors for each of the non CO2 gases, and then computing the $\Delta T_{RCP}85,_{GHG}$ as before using effCO2 rather than CO2 itself.

**We acknowledge the reviewer's excellent point. We have opted to keep the division into CO2 and non-CO2 because dividing into GHG and nonGHG components results in nonlinearities that violate the conditions of pattern**

**scaling.  The new Supplemental Section 1 provides more details as to why we made this choice.  We have also added a new paragraph of text in Section 4 that describes the above issues that the reviewer raises.**

The general formulation of the rest of the paper, and the treatment of the other two pattern scaling approaches, is broadly correct - but the presentation is often frustratingly vague. It is often not made clear what is in sample, and what is being tested. In Figure 8, are the same models being used to make the patterns and the test the errors? In Figure 11, is it 1pctCO2 or RCP85 being reconstructed?
The authors should correct the major errors above and restructure the paper to ensure concise and clear communication before resubmission.

**We acknowledge both of the items the reviewer points out.  We have clarified our description of Figure 8, also in line with a comment from Reviewer #1. For Figure 11, we have clarified what we are doing in the text.**

---

## Referee Report (RR1)

Review comments on the revised version of the manuscript

**Exploring precipitation pattern scaling methodologies and robustness among CMIP5 models**

by B. Kravitz, C. Lynch, C. Hartin and B. Bond-Lamberty

submitted to Geoscientific Model Development.

Recommendation: Accept with minor revisions.

In my opinion, there has been substantial progress in the paper, but publication in Geoscientific Model Development would still require some revisions.

**1   Specific comments**

1. P. 8, l. 18–31: You examine the non-CO2 pattern here, but the procedure of calculating that is only presented in subsection 4.2. (Note that the caption of Fig. 8 also refers to subsection 4.2.)

2. P. 9, l. 9: Being a small difference of two large quantities, $4P_{RCP8.5} - 3P_{CO2}$ is sensitive to noise occurring in these quantities. This has been mentioned previously, but this expression would elucidate the issue particularly well.

3. In many journals, sections and figures in the Supplement file are numbered as S1, S2, etc. Check the convention of the present journal. If numbering is changed, also remember to update references to these figures and sections, both in the main text and in the supplement.

4. Supplement, p. 2, l. 9–11: Please check the correctness of that sentence. It is Fig. 6 that uses the "wrong" $P$ while in Fig. 5 $\Delta T$ is extracted from the other group of models. If $\Delta T$ indeed is virtually similar for both groups, should the error be small just in Fig. 5?

5. I am not quite sure whether the discussion in section 4 of the Supplement is necessary. If the aim is to show that no universal non-CO2 pattern can be constructed, perhaps there would be more illuminating ways to show that. More detailed comments (only to be considered if this section will be retained):

    (a) P. 3, l. 25: Eq. (13) is a duplicate of (7). Perhaps it is not necessary to repeat, refer to (7).

    (b) It would be consistent to use the same amount of decimals in Eqs. (7) and (14). The coefficients would then take the form: 4.0, 3.0, 2.9, 1.9.

    (c) Eq. (20): state that this is only an assumption that you have employed (not an universal truth).

    (d) P. 4, l. 12: Does "the previous expression" refer to Eq. (18)?

    (e) I am afraid that there is an arithmetic error in deriving the equation given on p. 4, l. 12–13.

    (f) P. 4, l. 23: rationalize in more detail "which implies that"; moreover, specify what is the assumption that is violated.

**2   Minor comments**

1. P. 7, l. 14: Please specify whether "one method" refers here either to the epoch difference or regression method. If not, the text should be clarified.

2. P. 8, l. 2: Do "nonlinearities" refer to a nonlinearity of the response with respect to global mean temperature change?

3. P. 9, l. 16–19 (several occasions): Should $\overline{T}$ be replaced by $\Delta\overline{T}$? This issue also concerns the captions of Figs 8, 10, 11 and 12.

4. P. 9, l. 24–32: Here, it might be useful to refer to the rms differences given in Table 3.

5. Caption of Fig. 8: Should $\overline{T}$ be replaced by $\Delta\overline{T}$, and there be indices CO2 and non-CO2 for it?

6. Supplement, p. 3, l. 8: specify the years of "the latter part of the simulation".

---

## Author Response (AR2)

Response to reviewers

GMD-2016-258

Original reviewer comments in normal typeface.  **Responses in bold.**

Reviewer #1

P. 8, l. 18–31: You examine the non-CO2 pattern here, but the procedure of calculating that is only presented in subsection 4.2. (Note that the caption of Fig. 8 also refers to subsection 4.2.)

**Thanks for pointing that out.  We have moved that paragraph farther down in the manuscript**

**so it is after the description of non-CO2 pattern, and we have reordered the figures as**

**appropriate.**

P. 9, l. 9: Being a small difference of two large quantities, $4P_{RCP8.5} - 3P_{CO2}$ is sensitive to noise occurring in these quantities. This has been mentioned previously, but this expression would elucidate the issue particularly well.

**This is a good point, and we have now mentioned this explicitly in the manuscript.**

In many journals, sections and figures in the Supplement file are numbered as S1, S2, etc. Check the convention of the present journal. If numbering is changed, also remem- ber to update references to these figures and sections, both in the main text and in the supplement.

**We will confer with the editorial staff as to the best way to proceed.**

Supplement, p. 2, l. 9–11: Please check the correctness of that sentence. It is Fig. 6 that uses the

"wrong" P while in Fig. 5 ΔT is extracted from the other group of models. If ΔT indeed is virtually similar for both groups, should the error be small just in Fig. 5?

**You're right – we made an error in this sentence.  Thanks for catching that!**

I am not quite sure whether the discussion in section 4 of the Supplement is necessary. If the aim is to show that no universal non-CO2 pattern can be constructed, perhaps there would be more illuminating ways to show that.

**We included this section as part of our response to another reviewer, so we plan on retaining**

**it.**

More detailed comments (only to be considered if this section will be retained):

(a)  P. 3, l. 25: Eq. (13) is a duplicate of (7). Perhaps it is not necessary to repeat, refer to (7).

**We agree this is repetitive, but it repeats an equation in the main text, so we wanted to make**

**it easier on the reader (instead of flipping between two separate documents).**
(b)  It would be consistent to use the same amount of decimals in Eqs. (7) and (14). The
coefficients would then take the form: 4.0, 3.0, 2.9, 1.9.
**Agreed.  We have modified the manuscript accordingly.**
(c)  Eq. (20): state that this is only an assumption that you have employed (not an universal
truth).
**Good point.  Rephrased.**
(d)  P. 4, l. 12: Does "the previous expression" refer to Eq. (18)?
**Yes, corrected.**
(e)  I am afraid that there is an arithmetic error in deriving the equation given on p. 4, l. 12–13.
**Agreed.  Thanks for catching that.  Fortunately it doesn't change the conclusions.**
(f)  P. 4, l. 23: rationalize in more detail "which implies that"; moreover, specify what is the
assumption that is violated.
**Both points have now been clarified.**
P. 7, l. 14: Please specify whether "one method" refers here either to the epoch difference or
regression method. If not, the text should be clarified.
**We meant to refer to both methods.  Sorry for the confusion.**
P. 8, l. 2: Do "nonlinearities" refer to a nonlinearity of the response with respect to global mean
temperature change?
**This has been clarified in the text.**
P. 9, l. 16–19 (several occasions): Should T be replaced by $\Delta T$ ? This issue also concerns the
captions of Figs 8, 10, 11 and 12.
**Yes, thanks for catching that.**
P. 9, l. 24–32: Here, it might be useful to refer to the rms differences given in Table 3.
**Good idea.  Reference added.**

Caption of Fig. 8: Should T be replaced by ΔT , and there be indices CO2 and non-CO2 for it?

**We have replaced T by ΔT.  We are not sure where CO2 and non-CO2 indices might go in the**
**caption.**

Supplement, p. 3, l. 8: specify the years of "the latter part of the simulation".

**Description added.**

* * *

Reviewer #2

The paper is improved since the previous iteration. The exclusion of the 'physically based'
metric takes away the most serious issue from the previous draft.

**Thanks!**

However, I'm still left with some concerns. Firstly - some of the paper still seems to be devoted
into demonstrating the trivial - that the latter part of a 1pctCO2 simulation or RCP8.5
simulation can be used to reconstruct the same years from the same simulation. Why it would
be expected for any scaling approach to produce anything other than a near-perfect
reconstruction? As such, are these plots useful?

**We are not quite sure we understand the reviewer's comment.  It doesn't seem to be**
**consistent with what we believe we have demonstrated in the manuscript.  Clearly there is a**
**misunderstanding here, so we would appreciate specific suggestions as to where the**
**manuscript's clarity should be improved.**

Given that the paper is written as a best-practice guidance for pattern scaling precipitation, it
would be useful if the authors provided the reader more concise guidance in the abstract and in
the main text on best practice. The authors show that predicting RCP2.6 from RCP8.5 is
obviously stretching the approaches because the relative mix of forcings is so different at the
end of the 21st century. How different do forcing mixes have to be before the methods are no
longer useful (i.e. would it be acceptable to try and predict RCP4.5 from RCP8.5?). Which of the
two methods is better in what circumstance? How many models do you need to make your
pattern?

**The reviewer is asking excellent questions!  Per the instructions from the editor, we maintain**
**the focus of the manuscript as it presently is.  We hope that these sorts of questions, which**
**will be included in the online discussion, will spur follow-on studies (potentially conducted by**
**the present team) that explore these issues, which are at the heart of robustness of pattern**
**scaling.**

The assessment of inter-model prediction skill could potentially be useful, but it's under-
explored at present. The 'group 1' and 'group 2' distinctions seem arbitrary - and in many cases
contain models which are very closely related, meaning that they are questionable as
independent samples. It's not clear at present why most of the paper is conducted with only
one of these groups.
**The distinction is based on an assessment of model independence conducted by Lynch et al.**
**(submitted).  We realize we neglected to mention this, so we have now added a sentence**
**describing why Group 1 contains the models it does.**
This is a potentially rich topic, and in future the authors could ask some relevant questions -
like, for example, is there an optimal subset of models used to generate the pattern which
minimizes errors? i.e. if there is one really terrible model in there, are you better to leave it in
or take it out of your mean pattern?
**Great questions!  Per the instructions from the editor, we maintain the focus of the**
**manuscript as it presently is, but we acknowledge that these would be excellent future**
**investigations.**
It's interesting that the authors don't see the same relationship between number of models
used and skill in reconstruction for the two methods. Some analysis of why this is the case
would be useful.
**We agree.  Per the instructions from the editor, this sort of analysis is somewhat beyond**
**scope of the present manuscript, but we will keep it in mind for future assessments of**
**robustness.**
The non-CO2/CO2 separation makes more sense than in the previous iteration, but the
supplemental material requires some more clarification. In all RCPs, aerosols decline during the
21st century - if this is associated with a patterned precipitation response, then one would
never expect this aerosol-driven pattern to be a simple function of global mean temperature.
Rather, you would expect the pattern to scale with delta_T(non-GHG).
**We agree with the reviewer that the decomposition may be more complicated than is**
**presently allowed for under the definition of pattern scaling.  As this manuscript is focused**
**on an evaluation of pattern scaling methods, we are reluctant to substantially diverge from**
**those assumptions.  As we acknowledge in the manuscript, there are likely many instances**
**where pattern scaling does not work, although we do not go into great detail as to why that**
**may be.  It would be interesting to explore the sorts of things the reviewer suggests in future**
**work to further quantify the potential modes of failure of pattern scaling methods.**
By deriving the quantities by regressing delta_T(non-GHG) against delta_T(CO2), the authors
are hard-coding a fixed relationship between forcings. But, as the authors saw - there is no
need for these quantities to be linearly related. It should be noted that the relationship isn't

'quadratic' or 'exponential', it's just how the relative forcings happen to change in the RCP8.5
scenario over the 21st century, and because these relative forcings are very different in RCP2.6
to RCP8.5, the method produces large errors.
**We agree with the reviewer that these relationships are not necessarily linear, and we have**
**added a note in the manuscript to this effect in response to the following comment.**
It's also only fortuitous that delta_T(non-CO2) and delta_T(CO2) lie on a straight line in RCP8.5
(I suspect that this would not be true if the patterns were derived in RCP2.6). As such, it should
be noted that the equation for P_nonCO2 in the additional material is not general - it is tied to a
particular scenario.
**This is an excellent point.  We have added a note to this effect in the supplemental material**
**of the manuscript.**

[revised manuscript text omitted]
{pctCO2}}\bar{T}_{RCP8.5,CO_2}(116-140)$ $\Delta\hat{B} = P_{\mathrm{non-CO_2}}\Delta\bar{T}_{RCP8.5,non}$ and results are shown for $\hat{B} - B_{\mathrm{RCP8.5}}(116-140)$ $\Delta\hat{B} - \Delta B_{\mathrm{RCP8.5}}(116-140)$. (See Equation 1 and the discussion surrounding Equation 6 for details.)

[Figure]

**Figure 12.** As in Figure 3 but where $\hat{B} = P_{\mathrm{non-CO_2}}\bar{T}_{RCP2.6,nonCO_2}(227-251) + P_{\mathrm{1pctCO2}}\bar{T}_{RCP2.6,CO_2}(227-251)$  and results are shown for  $\Delta\hat{B} - \Delta B_{\mathrm{RCP2.6}}(227-251)$. (See Equation 1 and the discussion surrounding Equation 6 for details.)

**Exploring precipitation pattern scaling methodologies and robustness among CMIP5 models (Supplemental Online Material)**

Ben Kravitz[1], Cary Lynch[2], Corinne Hartin[2], and Ben Bond-Lamberty[2]

[1]Atmospheric Sciences and Global Change Division, Pacific Northwest National Laboratory, Richland, WA, USA.
[2]Joint Global Change Research Institute, Pacific Northwest National Laboratory, College Park, MD, USA.

*Correspondence to:* Ben Kravitz, P.O. Box 999, MSIN K9-30, Richland, WA 99352, USA. (ben.kravitz@pnnl.gov)

**1 Interpolation and Extrapolation in Time**

Here we reference Supplemental Figures 1-3. Discussion of these figures is included in Section 3.2 of the main paper.

**2 Comparison of Pattern Scaling Between Two Groups of Models**

Supplemental Figure 4 further supports the findings in Section 2 of the main paper by showing that the patterns $P(\mathbf{x})$ are not statistically different for Groups 1 and 2 except for isolated areas. Supplemental Figures 5 and 6 show differences in the reconstructions, averaged over years 116–140. More specifically, Supplemental Figure 5 shows differences $P_{\text{Group 2}}\Delta\bar{T}_{\text{Group 1}} - \Delta B_{\text{Group 2}}$  and Supplemental Figure 6 shows differences $P_{\text{Group 1}}\Delta\bar{T}_{\text{Group 2}} - \Delta B_{\text{Group 2}}$  .

The results in Supplemental Figures 5 and 6 have qualitatively more error than the results in Figure 3 in the main paper, but Supplemental Figure 5 has substantially more error than Supplemental Figure 6. This  shows that errors introduced by differences in $\Delta\bar{T}$ among the two groups are larger than errors introduced by differences in  $P$ among the two groups. As discussed in Section 3.2 in the main paper, practically no region is statistically significant for the regression and epoch difference methods in Supplemental Figures 5 and 6.

**3 Pattern Scaling for Non-CO$_2$ Forcings**

In Section 4 of the main portion of the paper, we discuss splitting the RCP8.5 scaling into a CO$_2$ portion and a non-CO$_2$ portion. We also discussed why we chose not to split the RCP8.5 scaling into a greenhouse gas and non-greenhouse gas portion. Here we provide more details on the rationale for that choice.

To perform this scaling, we begin with a restatement of Equation 1 in the main paper:

$$\Delta B(\mathbf{x},t) \approx \Delta\hat{B}(\mathbf{x},t) = P(\mathbf{x})\Delta\bar{T}(t) \tag{1}$$

where $P(\mathbf{x})$ describes a time-invariant spatial pattern (the spatial dimension is denoted by $\mathbf{x}$), and $\Delta\bar{T}(t)$ describes a time-varying (the time dimension is denoted by $t$) series of the change in global mean temperature, starting from a reference period $t = 0$ (often the preindustrial era).

In the first case, we split $\Delta B$ into a CO$_2$ portion and a non-CO$_2$ portion:

$$\Delta B_{\text{RCP8.5}} = P_{\text{RCP8.5}}\Delta T^{\text{RCP8.5}} = P_{\text{CO}_2}\Delta T^{\text{RCP8.5}}_{\text{CO}_2} + P_{\text{non−CO}_2}\Delta T^{\text{RCP8.5}}_{\text{non−CO}_2} \tag{2}$$

Solving for $P_{\mathrm{non-CO_2}}$ and assuming separability of temperature change into a $CO_2$ component and a non-$CO_2$ component, we obtain

$$P_{\mathrm{non-CO_2}} = \frac{\Delta B_{\mathrm{RCP8.5}} - P_{\mathrm{CO_2}}\Delta T_{\mathrm{CO_2}}^{\mathrm{RCP8.5}}}{\Delta T_{\mathrm{non-CO_2}}^{\mathrm{RCP8.5}}} \tag{3}$$

$$= \frac{P_{\mathrm{RCP8.5}}\Delta T^{\mathrm{RCP8.5}} - P_{\mathrm{CO_2}}\Delta T_{\mathrm{CO_2}}^{\mathrm{RCP8.5}}}{\Delta T_{\mathrm{non-CO_2}}^{\mathrm{RCP8.5}}} \tag{4}$$

$$= \frac{\Delta T_{\mathrm{CO_2}}^{\mathrm{RCP8.5}} + \Delta T_{\mathrm{non-CO_2}}^{\mathrm{RCP8.5}}}{\Delta T_{\mathrm{non-CO_2}}^{\mathrm{RCP8.5}}} P_{\mathrm{RCP8.5}} - \frac{\Delta T_{\mathrm{CO_2}}^{\mathrm{RCP8.5}}}{\Delta T_{\mathrm{non-CO_2}}^{\mathrm{RCP8.5}}} P_{\mathrm{CO_2}} \tag{5}$$

$$= (1+\beta)P_{\mathrm{RCP8.5}} - \beta P_{\mathrm{CO_2}} \tag{6}$$

where $\beta = \Delta T_{\mathrm{CO_2}}^{\mathrm{RCP8.5}}/\Delta T_{\mathrm{non-CO_2}}^{\mathrm{RCP8.5}}$. Because we want the pattern $P_{\mathrm{non-CO_2}}$ to be state-independent, we perform linear regression of $\Delta T_{\mathrm{CO_2}}^{\mathrm{RCP8.5}}$ against $\Delta T_{\mathrm{non-CO_2}}^{\mathrm{RCP8.5}}$ on the latter part (approximately last 50 years) of the simulation where the signal is clearest. We define the slope to be $\beta = 2.9588$; the fit has an $R^2$ value of 0.9883. Rounding, we obtain the expression

$$P_{\mathrm{non-CO_2}} = 4.0 P_{\mathrm{RCP8.5}} - 3.0 P_{\mathrm{CO_2}} \tag{7}$$

We note that this decomposition works for RCP8.5 because the relationship between $\Delta T_{\mathrm{CO_2}}^{\mathrm{RCP8.5}}$ and $\Delta T_{\mathrm{non-CO_2}}^{\mathrm{RCP8.5}}$ is approximately linear. This may not necessarily hold for other scenarios, such as RCP2.6, so a different methodology for recovering the non-$CO_2$ pattern may be necessary in that case.

In the second case, we split $\Delta B$ into a $CO_2$ portion, a non-$CO_2$ greenhouse gas portion (labeled "other GHG"), and a non-GHG portion:

$$\Delta B_{\mathrm{RCP8.5}} = P_{\mathrm{RCP8.5}}\Delta T^{\mathrm{RCP8.5}} = P_{\mathrm{CO_2}}\Delta T_{\mathrm{CO_2}}^{\mathrm{RCP8.5}} + P_{\mathrm{otherGHG}}\Delta T_{\mathrm{otherGHG}}^{\mathrm{RCP8.5}} + P_{\mathrm{non-GHG}}\Delta T_{\mathrm{non-GHG}}^{\mathrm{RCP8.5}} \tag{8}$$

Solving as above,

$$P_{\mathrm{non-GHG}} = \frac{\Delta B_{\mathrm{RCP8.5}} - P_{\mathrm{CO_2}}\Delta T_{\mathrm{CO_2}}^{\mathrm{RCP8.5}} - P_{\mathrm{otherGHG}}\Delta T_{\mathrm{otherGHG}}^{\mathrm{RCP8.5}}}{\Delta T_{\mathrm{non-GHG}}^{\mathrm{RCP8.5}}} \tag{9}$$

$$= \frac{P_{\mathrm{RCP8.5}}\Delta T^{\mathrm{RCP8.5}} - P_{\mathrm{CO_2}}\Delta T_{\mathrm{CO_2}}^{\mathrm{RCP8.5}} - P_{\mathrm{otherGHG}}\Delta T_{\mathrm{otherGHG}}^{\mathrm{RCP8.5}}}{\Delta T_{\mathrm{non-GHG}}^{\mathrm{RCP8.5}}} \tag{10}$$

$$= \frac{\Delta T_{\mathrm{CO_2}}^{\mathrm{RCP8.5}} + \Delta T_{\mathrm{otherGHG}}^{\mathrm{RCP8.5}} + \Delta T_{\mathrm{non-GHG}}^{\mathrm{RCP8.5}}}{\Delta T_{\mathrm{non-GHG}}^{\mathrm{RCP8.5}}} P_{\mathrm{RCP8.5}} - \frac{\Delta T_{\mathrm{CO_2}}^{\mathrm{RCP8.5}}}{\Delta T_{\mathrm{non-GHG}}^{\mathrm{RCP8.5}}} P_{\mathrm{CO_2}} - \frac{\Delta T_{\mathrm{otherGHG}}^{\mathrm{RCP8.5}}}{\Delta T_{\mathrm{non-GHG}}^{\mathrm{RCP8.5}}} P_{\mathrm{otherGHG}} \tag{11}$$

$$= (1+\gamma+\delta)P_{\mathrm{RCP8.5}} - \gamma P_{\mathrm{CO_2}} - \delta P_{\mathrm{otherGHG}} \tag{12}$$

where $\gamma = \Delta T_{\mathrm{CO_2}}^{\mathrm{RCP8.5}}/\Delta T_{\mathrm{non-GHG}}^{\mathrm{RCP8.5}}$ and $\delta = \Delta T_{\mathrm{otherGHG}}^{\mathrm{RCP8.5}}/\Delta T_{\mathrm{non-GHG}}^{\mathrm{RCP8.5}}$. At this point, the above procedure fails, because according to best-fits on the data plotted in Supplemental Figure 7, $\Delta T_{\mathrm{CO_2}}^{\mathrm{RCP8.5}}$ is approximately quadratic with $\Delta T_{\mathrm{non-GHG}}^{\mathrm{RCP8.5}}$, and $\Delta T_{\mathrm{otherGHG}}^{\mathrm{RCP8.5}}$ is approximately an exponential function of $\Delta T_{\mathrm{non-GHG}}^{\mathrm{RCP8.5}}$. Because neither of these relationships is linear, any derived patterns must be state-dependent if they are to be accurate.

**4 Pattern Scaling for RCP2.6**

In Supplemental Section 3, we derived the approximate equivalence

$$P_{\mathrm{non-CO_2}}^{\mathrm{RCP8.5}} = 4 P_{\mathrm{RCP8.5}} - 3 P_{\mathrm{CO_2}} \tag{13}$$

Using the same procedure, we can derive a similar equivalence for RCP2.6:

$$P_{\text{non}-\text{CO}_2}^{\text{RCP2.6}} = \underline{\sout{2.86}2.9}\,P_{\text{RCP8.5}} - \underline{\sout{1.86}1.9}\,P_{\text{CO}_2} \tag{14}$$

We note that the relationship between $\Delta T_{\text{CO}_2}^{\text{RCP2.6}}$ and $\Delta T_{\text{non}-\text{CO}_2}^{\text{RCP2.6}}$ is somewhat nonlinear, and regression revealed an $R^2$ value of 0.58.

We now discuss the implications of both of these formulations. First assume that $P_{\text{non}-\text{CO}_2}^{\text{RCP8.5}} = P_{\text{non}-\text{CO}_2}^{\text{RCP2.6}}$. Then

$$\Delta \hat{B}_{\text{RCP2.6}} = \qquad \Delta T_{\text{non}-\text{CO}_2}^{\text{RCP2.6}}(4P_{\text{RCP8.5}} - 3P_{\text{CO}_2}) + \Delta T_{\text{CO}_2}^{\text{RCP2.6}} P_{\text{CO}_2} \tag{15}$$

$$(\Delta T^{\text{RCP2.6}} - \Delta T_{\text{CO}_2}^{\text{RCP2.6}})(4P_{\text{RCP8.5}} - 3P_{\text{CO}_2}) + \Delta T_{\text{CO}_2}^{\text{RCP2.6}} P_{\text{CO}_2} \tag{16}$$

By definition, $\hat{B}_{\text{RCP2.6}} = \Delta T^{\text{RCP2.6}} P_{\text{RCP2.6}}$. Then

$$P_{\text{RCP2.6}} = \qquad \frac{(\Delta T^{\text{RCP2.6}} - \Delta T_{\text{CO}_2}^{\text{RCP2.6}})}{\Delta T^{\text{RCP2.6}}}(4P_{\text{RCP8.5}} - 3P_{\text{CO}_2}) + \frac{\Delta T_{\text{CO}_2}^{\text{RCP2.6}}}{\Delta T^{\text{RCP2.6}}} P_{\text{CO}_2} \tag{17}$$

$$(1-\beta)(4P_{\text{RCP8.5}} - 3P_{\text{CO}_2}) + \beta P_{\text{CO}_2} \tag{18}$$

$$= 4P_{\text{RCP8.5}} - 4\beta P_{\text{RCP8.5}} - 3P_{\text{CO}_2} + 4\beta P_{\text{CO}_2} \tag{19}$$

where $\beta = \Delta T_{\text{CO}_2}^{\text{RCP2.6}}/\Delta T^{\text{RCP2.6}} \approx 0.75$ by regression. Then

$$P_{\text{RCP2.6}} = P_{\text{RCP8.5}} \tag{20}$$

If we assume this equation to be true, then by Equation 18, $P_{\text{RCP8.5}} = (1-\beta)(4P_{\text{RCP8.5}} - $
$3P_{\text{CO}_2}) + \beta P_{\text{CO}_2}$. Then  $P_{\text{RCP8.5}} = P_{\text{CO}_2}$, which is clearly incorrect, invalidating the original assumption that $P_{\text{non}-\text{CO}_2}^{\text{RCP8.5}} = P_{\text{non}-\text{CO}_2}^{\text{RCP2.6}}$.

Evaluating the other expression , $P_{\text{non}-\text{CO}_2}^{\text{RCP2.6}} = 2.9 P_{\text{RCP8.5}} - 1.9 P_{\text{CO}_2}$,

$$\Delta \hat{B}_{\text{RCP2.6}} = \qquad \Delta T_{\text{non}-\text{CO}_2}^{\text{RCP2.6}}(\underline{\sout{2.86}2.9}\,P_{\text{RCP2.6}} - \underline{\sout{1.86}1.9}\,P_{\text{CO}_2}) + \Delta T_{\text{CO}_2}^{\text{RCP2.6}} P_{\text{CO}_2} \tag{21}$$

$$(\Delta T^{\text{RCP2.6}} - \Delta T_{\text{CO}_2}^{\text{RCP2.6}})(\underline{\sout{2.86}2.9}\,P_{\text{RCP2.6}} - \underline{\sout{1.86}1.9}\,P_{\text{CO}_2}) + \Delta T_{\text{CO}_2}^{\text{RCP2.6}} P_{\text{CO}_2} \tag{22}$$

By definition, $\hat{B}_{\text{RCP2.6}} = \Delta T^{\text{RCP2.6}} P_{\text{RCP2.6}}$. Then

$$\Delta T^{\text{RCP2.6}} P_{\text{RCP2.6}} - \underline{\sout{2.86}2.9}\,P_{\text{RCP2.6}}(\Delta T^{\text{RCP2.6}} - \Delta T_{\text{CO}_2}^{\text{RCP2.6}}) = \underline{\sout{1.86}1.9}\,P_{\text{CO}_2}(\Delta T^{\text{RCP2.6}} - \Delta T_{\text{CO}_2}^{\text{RCP2.6}}) + \Delta T_{\text{CO}_2}^{\text{RCP2.6}} P_{\text{CO}_2}$$
$$\tag{23}$$

Simplifying,

$$(P_{\text{RCP2.6}} - P_{\text{CO}_2})(\underline{\sout{2.86}2.9}\,\Delta T_{\text{CO}_2}^{\text{RCP2.6}} - \underline{\sout{1.86}1.9}\,\Delta T^{\text{RCP2.6}}) = 0 \tag{24}$$

So either $P_{\text{RCP2.6}} = P_{\text{CO}_2}$ or  $\Delta T_{\text{CO}_2}^{\text{RCP2.6}} = \frac{1.9}{2.9}\Delta T^{\text{RCP2.6}}$.

If the first condition is true, then $P_{\text{non}-\text{CO}_2}^{\text{RCP2.6}} = 0$, which  by Equation 14 implies that $P_{\text{RCP2.6}} = \frac{1.9}{2.9}P_{\text{CO}_2}$, violating the assumption that $P_{\text{RCP2.6}} = P_{\text{CO}_2}$. However, if the second condition is true, then by the above derivations, $\Delta T_{\mathrm{CO_2}}^{\mathrm{RCP2.6}}/\Delta T_{\mathrm{non-CO_2}}^{\mathrm{RCP2.6}} = 1.86$ $\Delta T_{\mathrm{CO_2}}^{\mathrm{RCP2.6}}/\Delta T_{\mathrm{non-CO_2}}^{\mathrm{RCP2.6}} = 1.9$, so through some simple algebra, $\Delta T_{\mathrm{CO_2}}^{\mathrm{RCP2.6}}/\Delta T^{\mathrm{RCP2.6}}$ $0.6504$. Performing regression on these two quantities yields a value of $0.7321$ with an $R^2$ value of $0.9610$. While this value is similar to the one derived above, it is sufficiently different to indicate a certain amount of nonlinearity for which pattern scaling cannot account.

[Figure]

**Figure 1.** Absolute values (left) of and differences (right) in the precipitation scaling pattern $P(\mathbf{x})$ (Equation 1) when different time periods are used to construct the pattern (years 1–50 versus years 116–140 of the 1pctCO2 simulation). Left column shows values of $P_{1-50}$, and right column shows values of $P_{1-50} - P_{116-140}$ (mm day$^{-1}$ K$^{-1}$). Values in subscripts denote that the associated quantities are calculated from an average over those years. Top row shows results for the regression method, and bottom row shows the epoch difference method. All values are calculated for a Group 1 multi-model average for the 1pctCO2 simulation. Stippling indicates a lack of statistical significance in the pattern of differences (Section 2.2).

[Figure]

**Figure 2.** As in Figure 3 in the main paper but where the reconstruction $\hat{B}$ is built on the pattern $P$ for years 1–50 (Group 1 average of the 1pctCO2 simulation), and global mean temperature $\Delta\bar{T}$ is averaged over years 116–140. That is, $\hat{B} = P_{1-50}(\mathbf{x})\Delta\bar{T}(116-140)$. Results shown are for the difference between the reconstruction and the actual model output $\hat{B} - B(\mathbf{x}, 116-140)$.

[Figure]

**Figure 3.** As in Figure 3 in the main paper but where the reconstruction $\hat{B}$ is built on the pattern $P$ for years 116–140 (Group 1 average of the 1pctCO2 simulation), and global mean temperature $\Delta\bar{T}$ is averaged over years 58–82. That is, $\hat{B} = P_{116-140}(\mathbf{x})\Delta\bar{T}(58-82)$. Results shown are for the difference between the reconstruction and the actual model output $\hat{B} - B(\mathbf{x}, 58-82)$.

[Figure]

**Figure 4.** Absolute values (left) of and differences (right) in time-invariant patterns $P(\mathbf{x})$ among the two groups of models (Table 1), calculated for the 1pctCO2 simulation. Left column shows the multi-model average for Group 2, and right column shows the differences in multi-model averages among the two groups. All values shown have units mm day$^{-1}$ K$^{-1}$. Stippling indicates a lack of statistical significance in the pattern of differences (Section 2.2).

[Figure]

**Figure 5.** As in Figure 3 in the main paper but where the reconstruction $\hat{B}$ is built on the pattern $P$ for Group 2 (average of years 116–140 of the 1pctCO2 simulation), and global mean temperature $\Delta\bar{T}$ is averaged over years 116–140 of Group 1. That is, $\hat{B} = P_{\text{Group2}}(\mathbf{x})\Delta\bar{T}_{\text{Group1}}(116-140)$. Results shown are for the difference between the reconstruction and the actual model output $\hat{B} - B_{\text{Group2}}(\mathbf{x}, 116-140)$.

[Figure]

**Figure 6.** As in Figure 3 in the main paper but where the reconstruction $\hat{B}$ is built on the pattern $P$ for Group 1 (average of years 116–140 of the 1pctCO2 simulation), and global mean temperature $\Delta\bar{T}$ is averaged over years 116–140 of Group 2. That is, $\hat{B} = P_{\text{Group1}}(\mathbf{x})\Delta\bar{T}_{\text{Group2}}(116-140)$. Results shown are for the difference between the reconstruction and the actual model output $\hat{B} - B_{\text{Group2}}(\mathbf{x}, 116-140)$.

[Figure]

**Figure 7.** As in Figure 9 of the main paper but with additional forcings, per the description in Supplemental Section 3. See main Section 4.2 and Supplemental Section 3 for further details on the quantities depicted here.